# Targeted degradation of aberrant tau in frontotemporal dementia patient-derived neuronal cell models

M Catarina Silva[1,2,3†], Fleur M Ferguson[4,5†], Quan Cai[4,5], Katherine A Donovan[4,5], Ghata Nandi[1,2,3], Debasis Patnaik[1,2,3], Tinghu Zhang[4,5], Hai-Tsang Huang[4,5], Diane E Lucente[6,7,8,9], Bradford C Dickerson[7,8,9], Timothy J Mitchison[10,11], Eric S Fischer[4,5], Nathanael S Gray[4,5]*, Stephen J Haggarty[1,2,3]*

[1]Chemical Neurobiology Laboratory, Center for Genomic Medicine, Massachusetts General Hospital, Harvard Medical School, Boston, United States; [2]Department of Neurology, Massachusetts General Hospital, Harvard Medical School, Boston, United States; [3]Department of Psychiatry, Massachusetts General Hospital, Harvard Medical School, Boston, United States; [4]Department of Cancer Biology, Dana-Farber Cancer Institute, Boston, United States; [5]Department of Biological Chemistry and Molecular Pharmacology, Harvard Medical School, Boston, United States; [6]Molecular Neurogenetics Unit, Center for Genomic Medicine, Massachusetts General Hospital, Harvard Medical School, Boston, United States; [7]MGH Frontotemporal Disorders Unit, Department of Neurology, Massachusetts General Hospital, Harvard Medical School, Charlestown, United States; [8]Gerontology Research Unit, Department of Neurology, Massachusetts General Hospital, Harvard Medical School, Charlestown, United States; [9]Alzheimer's Disease Research Center, Department of Neurology, Massachusetts General Hospital, Harvard Medical School, Charlestown, United States; [10]Department of Systems Biology, Harvard Medical School, Boston, United States; [11]Laboratory of Systems Pharmacology, Harvard Medical School, Boston, United States

*For correspondence:
nathanael_gray@dfci.harvard.edu (NSG);
shaggarty@mgh.harvard.edu (SJH)

†These authors contributed equally to this work

**Abstract** Tauopathies are neurodegenerative diseases characterized by aberrant forms of tau protein accumulation leading to neuronal death in focal brain areas. Positron emission tomography (PET) tracers that bind to pathological tau are used in diagnosis, but there are no current therapies to eliminate these tau species. We employed targeted protein degradation technology to convert a tau PET-probe into a functional degrader of pathogenic tau. The hetero-bifunctional molecule QC-01–175 was designed to engage both tau and Cereblon (CRBN), a substrate-receptor for the E3-ubiquitin ligase CRL4[CRBN], to trigger tau ubiquitination and proteasomal degradation. QC-01–175 effected clearance of tau in frontotemporal dementia (FTD) patient-derived neuronal cell models, with minimal effect on tau from neurons of healthy controls, indicating specificity for disease-relevant forms. QC-01–175 also rescued stress vulnerability in FTD neurons, phenocopying CRISPR-mediated *MAPT*-knockout. This work demonstrates that aberrant tau in FTD patient-derived neurons is amenable to targeted degradation, representing an important advance for therapeutics.
DOI: https://doi.org/10.7554/eLife.45457.001

## Introduction

Tauopathies, such as frontotemporal dementia (FTD), progressive supranuclear palsy (PSP) and Alzheimer's disease (AD), are a group of neurodegenerative diseases characterized by the pathological

accumulation of hyper-phosphorylated tau (P-tau) protein, in the form of intracellular paired helical filaments (PHFs) or neurofibrillary tangles (NFTs), within neurons and glia of affected brain regions, leading to cell death (*Kosik et al., 1989*; *Morris et al., 2011*; *Cruts and Van Broeckhoven, 2015*; *Ghetti et al., 2015*; *Neumann et al., 2015*; *Olney et al., 2017*; *Goedert, 2004*). Tauopathies can be either sporadic or inherited as autosomal dominant disease when caused by mutations in the *MAPT* gene encoding the microtubule-associated protein tau. FTD is the most common form of dementia in individuals under 60 years of age, affecting approximately 60,000 individuals in the USA alone, with an economic burden that is nearly twice that reported for AD (*Galvin et al., 2017*). Despite its devastating effects, there are currently no effective disease-modifying therapies, highlighting an urgent unmet need.

One of the major bottlenecks in developing effective therapies for tauopathies resides in the fact that molecular mechanisms leading to neuronal toxicity and death are still not entirely understood (*Congdon and Sigurdsson, 2018*; *Panza et al., 2016*; *Medina, 2018*; *Götz et al., 2013*). Cumulative evidence from murine tauopathy models and postmortem patient brain studies suggests that early tau post-translational modifications lead to misfolding, mislocalization, oligomerization, and changes in solubility. These events appear to be determinant toxicity effectors (*Johnson and Stoothoff, 2004*; *Min et al., 2015*; *Wang et al., 2009*; *Götz et al., 2013*; *Kopeikina et al., 2012*; *Tian et al., 2013*; *Yanamandra et al., 2013*; *Cowan and Mudher, 2013*), whereas tau tangles alone are not sufficient to cause neuronal death (*Cowan and Mudher, 2013*; *de Calignon et al., 2010*; *Kopeikina et al., 2012*; *Santacruz et al., 2005*; *Spires et al., 2006*). Therefore, targeting *early* forms of toxic tau for clearance may facilitate the study of their role in disease etiology and be a promising therapeutic strategy to reduce neuronal degeneration.

A challenge in developing cell-permeable small molecules that target tau is the lack of a well-defined tau fold and active sites, in disease. Current investigative tau-directed therapeutics have focused on aggregation inhibitors (*Brunden et al., 2010*; *Panza et al., 2016*; *Bulic et al., 2009*), activators of protein clearance through autophagy (*Boland et al., 2008*; *Krüger et al., 2012*; *Medina, 2018*; *Wang and Mandelkow, 2012*; *Rubinsztein et al., 2015*), and inhibition of tau kinases (*Dolan and Johnson, 2010*; *Medina, 2018*). Moreover, anti-tau immunotherapy has shown promise in animal models, but antibody affinity and specificity as well as strong immune responses pose critical challenges (*Gu et al., 2013*; *Panza et al., 2016*; *Pedersen and Sigurdsson, 2015*; *Novak et al., 2017*; *Yanamandra et al., 2013*). An alternative and promising strategy has focused on using anti-sense oligonucleotides (ASO) to decrease tau expression, leading to reversal of tau pathology in mouse and non-human primate models (*DeVos et al., 2017*; *Mignon et al., 2018*; *Xu et al., 2014*). Still, given potential limitations with existing approaches, developing small molecule agents that target *early* forms of toxic tau, and result in their degradation, may represent a uniquely advantageous strategy for halting tauopathies.

Targeted protein degradation approaches have expanded the landscape of druggable proteins by providing a mechanism to transform a non-functional protein binder into an effective targeted degrader (*Gechijian et al., 2018*). Targeted protein degraders, also known as PROTACs (PROteolysis TArgeting Chimeras), are an emerging strategy for ablating previously undruggable protein functions (*Cromm et al., 2018*; *Gechijian et al., 2018*; *Xie et al., 2014*; *Lai and Crews, 2017*). Structurally, degraders are hetero-bivalent compounds where a small molecule binder of the protein of interest is linked via a short linker to an E3-ligase recruiting molecule, such as the CRBN binder thalidomide (*Figure 1A*). This results in ternary complex formation between the protein of interest, the degrader molecule and the E3-ligase complex, in this case CUL4-RBX1-DDB1-CRBN (CRL4[CRBN]), and induces ubiquitination and subsequent proteasomal degradation of the protein of interest (*Figure 1B*) (*Fischer et al., 2014*; *Krönke et al., 2014*; *Lu et al., 2014*; *Chamberlain et al., 2014*; *Collins et al., 2017*; *Metzger et al., 2014*).

In this study, we harnessed targeted protein degradation technology to transform one of the most clinically advanced tau PET (positron emission tomography) tracers, [18]F-T807 (or [18]F-AV-1451), into the tau degrader QC-01–175 (*Chien et al., 2013*; *Xia et al., 2013*; *Lowe et al., 2016*). [18]F-T807 binds P-tau in vivo in a conformation-dependent manner, with highest efficiency in *MAPT* mutation carriers that produce AD-like PHF pathology (*Jones et al., 2018*), showing increased uptake in brain regions with significant tau deposition burden and little background binding in the cortex of normal brains. [18]F-T807 is the most advanced tau PET tracer in terms of its investigation in vivo, with studies completed in numerous tauopathies including AD and PSP (*Holt et al., 2016*; *Jones et al., 2018*;

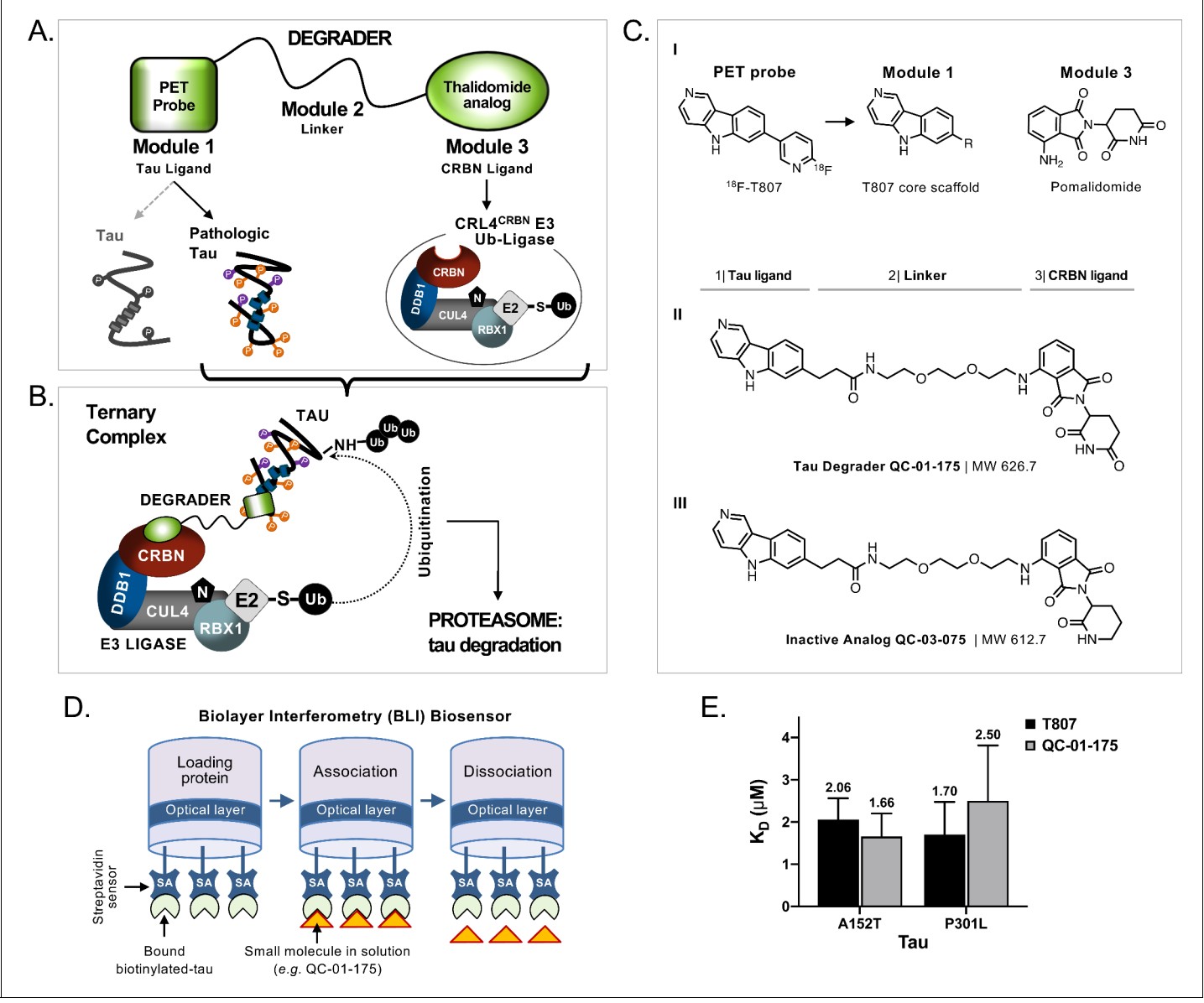

**Figure 1.** Design and working model for a new hetero-bifunctional tau degrader. (**A**) A degrader molecule was designed to preferentially recognize disease-associated forms of tau (Module 1), and simultaneously engage with CRBN in the CRL4$^{CRBN}$ E3 ubiquitin ligase complex (Module 3). (**B**) Degrader-mediated association of tau with CRL4$^{CRBN}$ and formation of a ternary complex is predicted to mediate tau ubiquitination and degradation by the proteasome. (**C**) QC-01–175 (**I**) was synthesized based on the T807 core scaffold for tau recognition, a thalidomide analog E3 ligand (pomalidomide) for CRBN engagement, and a linker designed for maximum target clearance efficiency (**II**). QC-03–075 is the inactive analog (**III**). (**D**) BLI Streptavidin (SA) biosensor assay to measure recombinant tau protein affinity to small molecules (e.g. T807, QC-01–175). (**E**) BLI results indicate that, in vitro, QC-01–175 binds to variant forms of tau within the same order of magnitude as T807. Bars represent mean $K_D$ (μM)±SD (n ≥ 3). *Figure 1—figure supplement 1* shows representative BLI sensograms and steady-state graphs for tau WT and each variant affinity to QC-01–175 and control compounds, with respective $K_D$ values. *Figure 1—figure supplement 2* shows QC-01–175 effect on monoamine oxidase (MAO) activity. The following figure supplements are available for *Figure 1*.

DOI: https://doi.org/10.7554/eLife.45457.002

The following figure supplements are available for figure 1:

**Figure supplement 1.** In vitro characterization of tau-binding affinity to QC-01–175 and control compounds.

DOI: https://doi.org/10.7554/eLife.45457.003

**Figure supplement 2.** QC-01–175 effect on MAO activity.

DOI: https://doi.org/10.7554/eLife.45457.004

*Murugan et al., 2018*; *Saint-Aubert et al., 2017*; *Schöll et al., 2016*; *Smith et al., 2016*; *Spina et al., 2017*; *Johnson et al., 2016*; *Schonhaut et al., 2017*; *Smith et al., 2018*). We now demonstrate that a T807-derived degrader molecule, QC-01–175, preferentially degrades tau species in FTD patient-derived neuronal cell models, while sparing tau in healthy controls. Our results suggest that this degrader strategy may offer a promising opportunity for neutralizing the neurotoxic effects of tau.

## Results

### Design and in vitro testing of targeted tau degraders

We designed and synthesized a small library of 25 hetero-bifunctional molecules and candidate tau degraders, containing variants of the tau PET tracer T807 (module 1), coupled through a linker of variable size and attachment chemistry (module 2) to the E3-ligase recruiting ligand pomalidomide, which engages with CRL4$^{CRBN}$ (module 3, *Figure 1A and C–I*). Through iterative design and testing in vitro and in ex vivo human neuronal cells, where each module element played a role in tau degradation efficacy, we identified the lead compound QC-01–175 (*Figure 1C–II*) (detailed structure-activity-relationship analysis to be published elsewhere). This compound contains the core 5*H*-pyrido[4,3-*b*]indole scaffold of T807, with the fluoro-pyridine ring replaced by a [PEG]$_2$ linker attached to pomalidomide. We generated the corresponding negative control compound, QC-03–075, by replacing the glutarimide in pomalidomide with a δ-lactam ring, thus abrogating CRBN-binding capacity (*Figure 1C–III*) (*Huang et al., 2018*).

A biolayer interferometry (BLI) assay was developed to measure in vitro affinity of the degrader molecule against wild-type (WT) and two variant forms of recombinant human tau: A152T and P301L (*Figure 1D*) (*Kumaraswamy and Tobias, 2015*; *Petersen, 2017*; *Shah and Duncan, 2014*). Known tau binders T807 and PE859 were used as positive controls to validate the BLI biosensor (*Figure 1—figure supplement 1A–B*) (*Okuda et al., 2017*; *Okuda et al., 2015*). In this in vitro format, although weaker binding was observed compared to PE859 (*Figure 1—figure supplement 1A*), QC-01–175 was able to bind to immobilized, soluble forms of WT ($K_D$1.2 μM), A152T ($K_D$1.7 μM) and P301L ($K_D$2.5 μM) within the same order of magnitude as the control T807 ($K_D$1.8 μM, 2.1 μM and 1.7 μM, respectively) (*Figure 1E*, *Figure 1—figure supplement 1A–B III, VI*). As expected, in vitro QC-01–175 and T807 displayed a significantly lower affinity for all recombinant immobilized tau forms than the reported $K_D$ of T807 for native, aggregated tau filaments from AD brain sections ($K_D$15 nM) (*Chien et al., 2013*; *Gobbi et al., 2017*; *Xia et al., 2013*). This property enables T807-based PET tracers to distinguish pathological tau species from the functional tau found in cells, and suggested that QC-01–175 may exhibit similar conformational selectivity. Comparison between binding of recombinant WT protein and variant tau forms, to either QC-01–175 or the controls PE859 and T807, indicates that T807 and derivatives have low affinity for monomeric, soluble protein, independent of the presence of a tau variant (*Figure 1—figure supplement 1B*).

The PET tracer $^{18}$F-T807 has off-target activity against monoamine oxidase-B (MAO-B) and monoamine oxidase-A (MAO-A) (*Lemoine et al., 2018*; *Vermeiren et al., 2018*). MAO inhibition interferes with neurotransmitter signaling, complicating phenotypic analysis, and in the clinic is associated with undesired risks of hypertensive crisis and with drug-drug interactions. To test if modifications introduced to the T807 scaffold increase MAO off-target activity of the degrader, we tested T807 and QC-01–175 in an in vitro MAO assay, using parnate, a known MAO inhibitor as positive control, and pomalidomide as a negative control. QC-01–175 showed reduced inhibition of MAO relative to T807 ($IC_{50}$8.56 μM vs 0.14 μM, respectively) (*Figure 1—figure supplement 2*). Together, in vitro testing of QC-01–175 suggests that it has a preserved tau ligand-binding capacity and reduced off-target MAO inhibition.

### QC-01–175 promotes tau clearance in a human neuronal cell model of tauopathy

We employed two tauopathy cell models to assess the activity of QC-01–175 in human neurons, one derived from a PSP patient carrier of the tau-A152T risk variant, and one derived from a behavioral-variant FTD patient carrier of the tau-P301L autosomal dominant mutation (both heterozygous, *Figure 2—source data 1*) (*Seo et al., 2017*; *Silva et al., 2016*; *Coppola et al., 2012*; *Mirra et al.,*

*1999*). Post-mitotic neurons were generated from patients' induced pluripotent stem cells (iPSC)-derived cortical-enriched neural progenitor cells (NPCs) subsequently differentiated into neurons for a period of 6 to 8 weeks, as previously described (*Cheng et al., 2017*; *Silva et al., 2016*; *Seo et al., 2017*). These patient-specific cellular models that express tau at endogenous levels, recapitulate disease-relevant tau phenotypes ex vivo (*Fong et al., 2013*; *Silva et al., 2016*) and present a valuable opportunity to test emerging therapeutics directly against a clinically relevant model of human disease (*Dolmetsch and Geschwind, 2011*; *Silva et al., 2017*; *Wan et al., 2015*; *Inoue et al., 2014*).

In A152T neurons treated with QC-01–175 for 24 hr (*Figure 2A*), we observed a concentration-dependent reduction in total tau and P-tau levels by western blotting analysis (*Figure 2B*) and by tau ELISAs (*Figure 2C*). P-tau$^{S396}$ and high-molecular-weight P-tau$^{S396}$, identified as SDS-insoluble bands > 250 kDa (*Figure 2B*), were effectively degraded at concentrations as low as 100 nM QC-01–175, whereas maximal total tau clearance ($D_{max, 24h}$) was achieved at 10 µM, consistent with expected binding preference of a T807-derivative for insoluble P-tau (*Chien et al., 2013*; *Lowe et al., 2016*; *Marquié et al., 2015*; *Xia et al., 2013*). The observed $D_{max, 24h}$ of 75% for total tau and of 85% for P-tau$^{S396}$, averaged across independent replicates (*Figure 2—figure supplement 1*), was well above the 50% mutant tau allele expressed by heterozygous neurons, indicating that both variant and WT forms of tau are recognized and degraded. Previous mass spectrometry (MS) FLEXITau analysis demonstrated that tau-A152T and tau-WT accumulate in similar proportions in these neurons (*Silva et al., 2016*), suggesting that QC-01–175 targets pathogenic tau conformations that include both WT and A152T tau, rather than tau variant exclusively. As the tau species present in iPSC-derived neurons vary in a differentiation- and maturation-dependent manner, we observed some variability in degradation effect across biological replicates (*Figure 2—figure supplement 1A–I*). To ascertain this inherent variability, we plotted the levels of total tau and P-tau$^{S396}$ upon 24 hr treatment with 1 µM or 10 µM doses of QC-01–175 across nine independent experiments. QC-01–175 promoted between 50 and 100% tau clearance, a statistically significant reduction relative to vehicle (*Figure 2—figure supplement 1J*). Conversely, the negative control QC-03–075 (*Figure 2D*) showed no effect on the levels of total tau and P-tau$^{S396}$ at all concentrations tested, indicating that engagement with CRL4$^{CRBN}$ is required for QC-01–175-mediated tau clearance (*Figure 2E,F* and *Figure 2—figure supplement 1J*).

We employed immunofluorescence (IF) of tau in A152T neurons to assess the effect of treatment with 10 µM QC-01–175 for 24 hr. Total tau (K9JA antibody), P-tau$^{S396/S404}$ (PHF-1 antibody) and neuronal marker MAP2 were imaged, and a clear reduction in tau and P-tau$^{S396/S404}$ were observed relative to vehicle- or QC-03–075-treated neurons (*Figure 2G*).

Next, we tested whether QC-01–175 could degrade multiple FTD-relevant tau species by repeating the ELISA tau quantification assay in tau-P301L neurons. QC-01–175 effected concentration-dependent degradation of total and P-tau$^{S396}$ at the 24 hr time point, with a $D_{max, 24h}$ of 60% achieved at 1 µM compound for both tau species (*Figure 2—figure supplement 2A*). QC-03–075 had minimal effect on tau levels (*Figure 2—figure supplement 2B*). These effects were corroborated by western blotting analysis, where 24 hr treatment with 1 µM QC-01–175 resulted in a 70% reduction in total tau and an 80% reduction of P-tau$^{S396}$, comparable to the effect in A152T neurons, whereas QC-03–075 had no effect on the levels of either tau species (*Figure 3—figure supplement 1A,B*). Collectively, these results demonstrate that QC-01–175 targets multiple forms of tau for degradation in FTD neuronal cell models.

## QC-01–175 preferentially degrades tau species found in FTD neurons vs. healthy controls

To investigate the ability of QC-01–175 to discriminate between normal and disease-associated tau, we tested our degrader against three independent, non-affected iPSC-derived neuronal cell models expressing WT tau, control neurons 1–3, in a tau ELISA assay (*Figure 2—source data 1*). These control neuronal cultures also contain reduced levels of the low-solubility P-tau$^{S396}$ species present in the FTD-patient-derived neurons (*Figure 3—figure supplement 1C vs. A, B*) (*Silva et al., 2016*). No significant degradation of tau or P-tau$^{S396}$ was observed after 24 hr treatment with either QC-01–175 or negative control QC-03–075 at concentrations up to 1 µM in all three control neurons (*Figure 2H–J*). We corroborated these results by western blotting, where 1 µM QC-01–175 was unable to induce significant degradation of tau or P-tau after 24 hr in control line 1 (*Figure 3—figure supplement 1C*). At the highest dose of 10 µM, QC-01–175 induced ~20% degradation of total

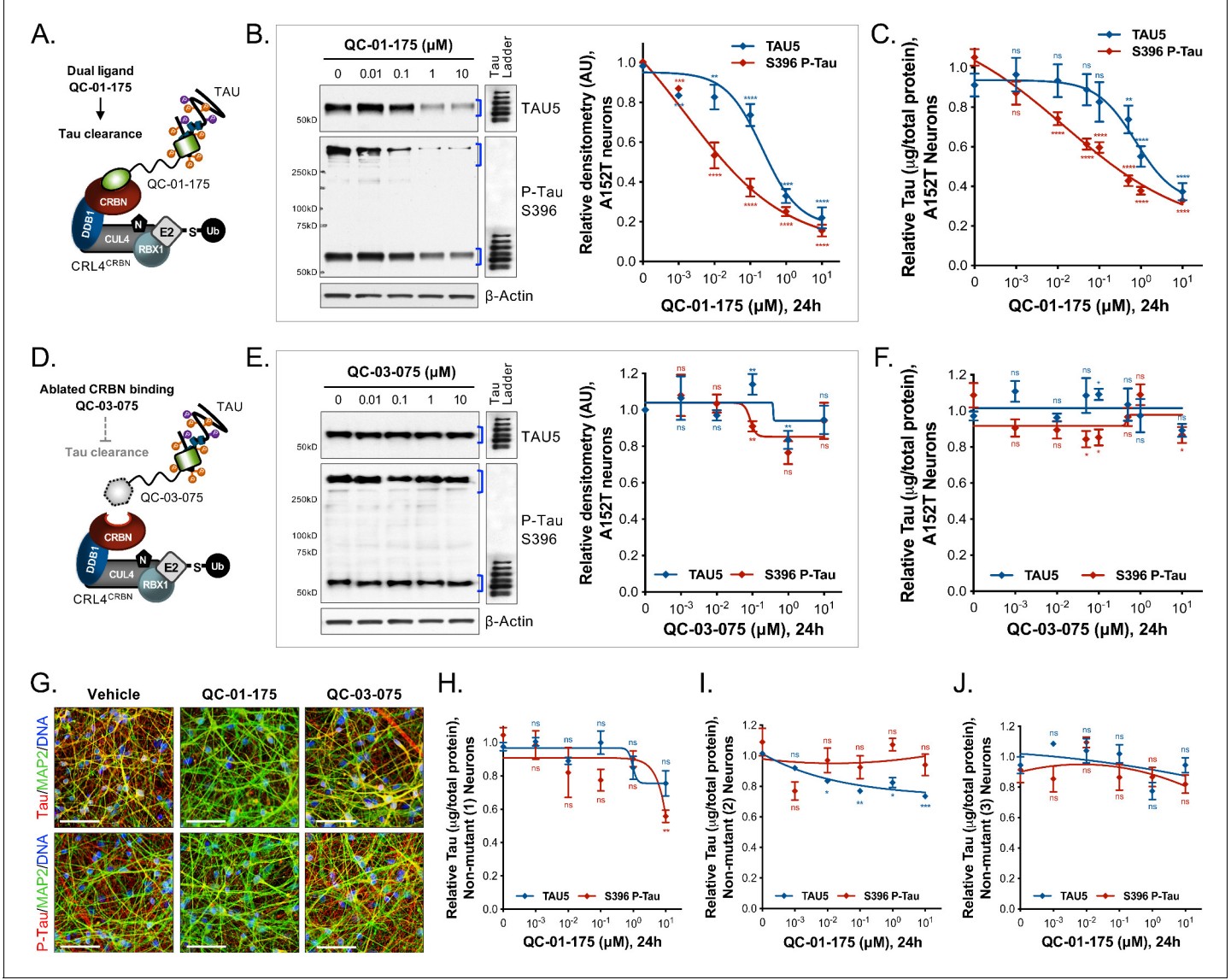

**Figure 2.** Concentration effect of QC-01–175 (**A**) on tau protein levels of A152T and control neurons. Analysis of total tau (TAU5) and phospho-tau (S396 P-tau) levels upon treatment by western blot (**B**) and ELISA (**C**). Analysis of total tau (TAU5) and phospho-tau (S396 P-tau) levels upon treatment with the negative control QC-03–075 (**D**), by western blot (**E**) and ELISA (**F**). Representative western blots are shown (**B, E**) with mean densitometry quantification (bands corresponding to brackets)±SEM (*n* = 3). (**C, F**) For ELISA, data points represent mean tau levels (µg of total protein) normalized to vehicle-treated ± SEM (*n* = 4). Both assays show QC-01–175 dose-dependent effect on tau levels, with QC-03–075 minimal effect (~10%). (**G**) IF of A152T neurons treated with vehicle or 10 µM compound, immuno-probed for total tau (K9JA, red), P-tau (PHF-1, red) and the neuronal marker MAP2 (green), scale bar 50 µm. (**H–J**) Tau ELISA of control neurons treated with QC-01–175, which did not show a dose-dependent effect. (**H**) 8330–8-RC1 line; (**I**) MGH2069-RC1 line; (**J**) CTR2-L17-RC2 line. Data points represent mean tau levels (µg of total protein) normalized to vehicle-treated ± SEM (*n* = 3). All neurons were differentiated for 6 weeks and treated with compound for 24 hr. Student T-test between each concentration and vehicle-treated tau levels [ns]$p > 0.05$, *$p < 0.05$, **$p < 0.01$, ***$p < 0.001$, ****$p < 0.0001$. *Figure 2—figure supplement 1* depicts the variability of degrader effect across biological replicates, by western blot, with an overall 60% to 90% efficacy. *Figure 2—figure supplement 2* shows degrader effect in P301L neurons and compares concentration effect across all lines. *Figure 2—source data 1* summarizes the information pertaining to each cell line included in this study. *Figure 2—source data 2* includes all ELISA data. The following figure supplements are available for *Figure 2*.

DOI: https://doi.org/10.7554/eLife.45457.005

The following source data and figure supplements are available for figure 2:

**Source data 1.** Human neural cell lines derived from tauopathy-affected (progressive supranuclear palsy, PSP or behavioral variant of FTD, bvFTD) and age-matched unaffected individuals, and *MAPT* KO line employed in this study.

DOI: https://doi.org/10.7554/eLife.45457.008

**Source data 2.** Numerical description and statistics for data presented in *Figure 2* and respective supplement 2 ELISAs.

*Figure 2 continued on next page*

*Figure 2 continued*

DOI: https://doi.org/10.7554/eLife.45457.009

**Figure supplement 1.** Variability of the effect of QC-01–175 across biological replicates.

DOI: https://doi.org/10.7554/eLife.45457.006

**Figure supplement 2.** Demonstration of QC-01–175 effect in tau-P301L neurons.

DOI: https://doi.org/10.7554/eLife.45457.007

tau in two of the control lines (1 and 2) and had no effect in the third line. Similarly, 10 µM QC-01–175 degraded ~40% of P-tau$^{S396}$ in control line 1 and had no significant effect in control lines 2 and 3. Comparative analysis of the effect of QC-01–175 across all cell lines tested suggests that QC-01–175 is significantly more effective at inducing degradation of the forms of tau present in FTD patient-derived neurons (*Figure 2—figure supplement 2C–E*).

## Induced tau degradation is proteasome-dependent and occurs via CRL4$^{CRBN}$ binding

To examine the mechanism of action of QC-01–175, we targeted each component predicted to be required for degrader-mediated tau clearance (*Figure 3A–C*, *top panels*). We treated A152T neurons with QC-01–175 at the minimum dose for maximum tau clearance, that is 1 µM for 24 hr (50–100% clearance), with or without pre-treatment for 6 hr with specific inhibitors, followed by western blotting analysis of total tau and P-tau$^{S396}$. First, excess lenalidomide or T807 that saturate ligand binding sites of CRBN or tau, respectively (*Figure 3A*), reversed QC-01–175-mediated tau degradation in a concentration-dependent manner (*Figure 3A,D*). Next, pre-treatment with MLN4924 (Pevonedistat), which inhibits NAE (neddylation activating enzyme NEDD8, and therefore neddylation 'N' of CUL4) (*Lan et al., 2016*) and consequently E3 ligase activity (*Figure 3B*), or pre-treatment with the irreversible proteasome inhibitor carfilzomib (*Figure 3C*) (*Huang and Dixit, 2016*), caused a concentration-dependent rescue of the effect of QC-01–175 (*Figure 3B,C,E,F*). In contrast, the autophagy inhibitor bafilomycin A1 (Baf.A1) did not reverse the degrader effect and further promoted tau clearance, possibly as a compensatory response between autophagy and proteasome clearance pathways (*Figure 3B,E*). The negative control QC-03–075 showed no significant effect on the levels of tau (*Figure 3A,B,D,E*). All inhibitors were utilized at concentrations that did not affect neuronal viability (*data not shown*). We also confirmed that QC-01–175 did not significantly affect the levels of CRBN by 24 hr, which remained constant across all concentrations (*Figure 3—figure supplement 1D*). Taken together, these data establish that QC-01–175-mediated tau degradation is dependent on CRBN and tau binding, neddylation (E3 ligase function) and proteasome function, but not autophagy, consistent with the proposed mechanism of action (*Figure 1B*).

As additional controls for specificity, we treated all neurons with the CRBN ligand lenalidomide or the tau ligand T807 at concentrations of 1 µM and 10 µM (*Figure 2—figure supplement 1J*, *Figure 3—figure supplement 1A–C*,). Across biological replicates, lenalidomide induced a significant up-regulation of P-tau$^{S396}$ (~30% and 90% at 1 µM and 10 µM, respectively; *Figure 2—figure supplement 1J*) in A152T neurons, indicating that CRBN inhibition alone may lead to accumulation of P-tau in these neurons, which are more sensitive to modulators of the proteasome than healthy neurons (*Silva et al., 2016*). We also observed that T807 treatment alone could lead to a small upregulation of both total and P-tau (10–30%) in A152T neurons (*Figure 2—figure supplement 1J*) *Jones et al., 2018*. Together, these results show that the action of the bifunctional degrader QC-01–175 cannot be recapitulated by treatment with its independent components.

Next, we performed co-immunoprecipitation (co-IP) assays in degrader-treated A152T neurons, by pulling down either the CRL4$^{CRBN}$ component DDB1 or tau, to test ternary complex predicted interactions (*Figure 1B*). We performed this experiment upon 4 hr of 1 µM QC-01–175 treatment, under the assumption that maximum engagement would be captured within a shorter interval. To ensure detection of the tau-degrader-CRL4$^{CRBN}$ binding complex before tau clearance occurred, we pre-treated samples with either carfilzomib or bortezomib proteasome inhibitors for 30 min (*Huang and Dixit, 2016*) to promote accumulation of the complex. Immunoprecipitation of DDB1 and follow-up western blotting analysis showed co-IP with both total tau (TAU5) and P-tau (PHF-1) upon QC-01–175 treatment ±carfilzomib (*Figure 4A*). There was also a weak detection of total tau and P-tau in vehicle and negative control QC-03–075 treated samples, suggesting that some basal

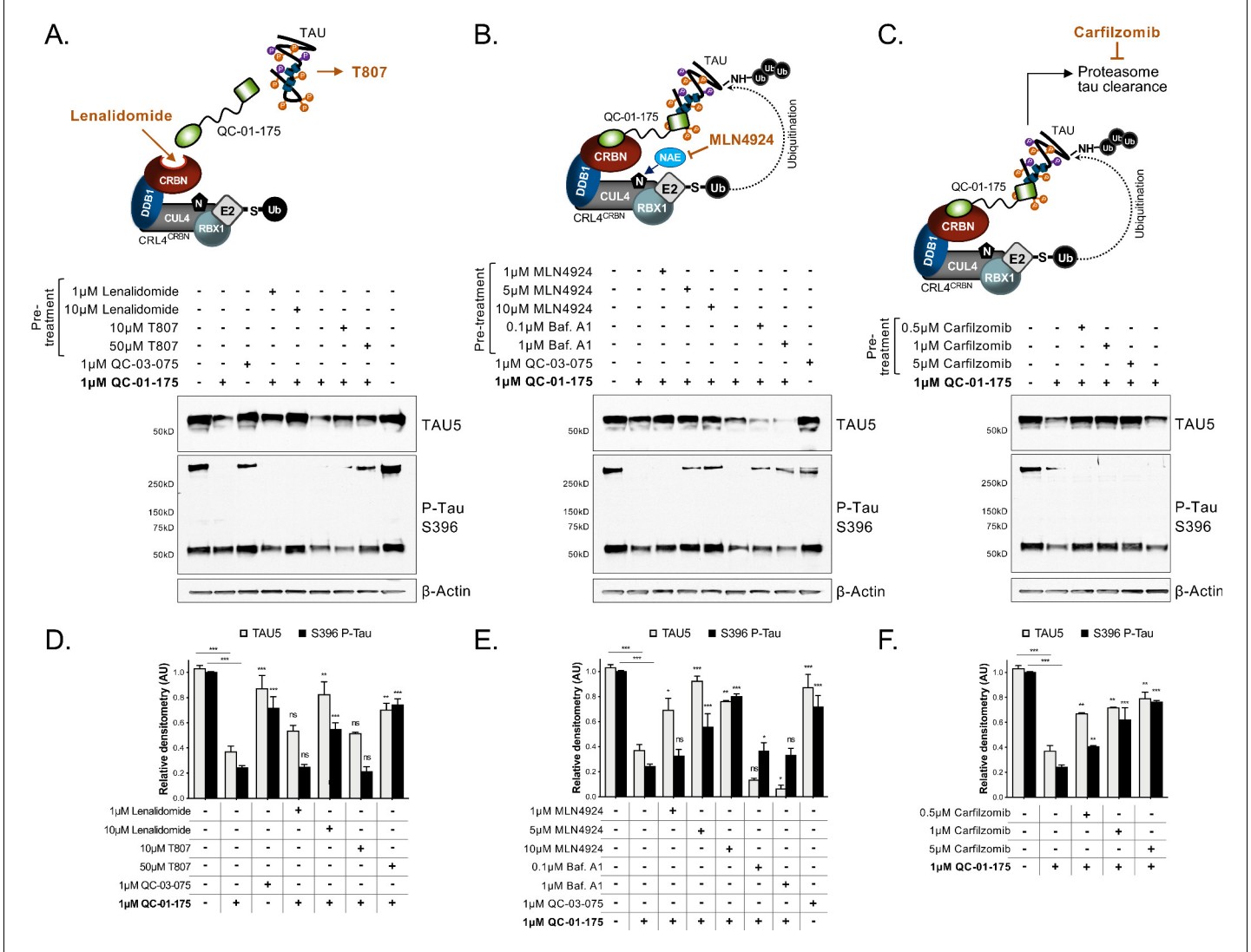

**Figure 3.** Mechanism of QC-01–175 clearance of tau is CRL4$^{CRBN}$ and UPS-dependent. Neurons were pre-treated for 6 hr with (**A**) either CRBN ligand excess lenalidomide or tau ligand excess T807, (**B**) the NAE inhibitor MLN4924, the autophagy inhibitor Baf.A1, or (**C**) the proteasome inhibitor carfilzomib; followed by 18 hr treatment with QC-01–175 (or negative control QC-03–075), for a total of 24 hr. Total (TAU5) and P-tau S396 levels were analyzed by western blotting. (**A–C**) Representative blots are shown. (**D–F**) Densitometry bars represent tau mean intensity values ± SD (*n* = 3), relative to vehicle-treated samples. Student T-test of QC-01–175 samples relative to vehicle treated, and the remainder bars show p-value of each pre-treatment relative to QC-01–175 to assess rescue of clearance effect (***p<0.001, **p<0.01, *p<0.05, $^{ns}$p>0.05). A152T neurons were differentiated for 6 weeks. *Figure 3—figure supplement 1* includes additional specificity controls for A152T, P301L and control neuronal models. The following figure supplement is available for *Figure 3*.

DOI: https://doi.org/10.7554/eLife.45457.010

The following figure supplement is available for figure 3:

**Figure supplement 1.** Additional specificity controls for QC-01–175-mediated tau clearance.

DOI: https://doi.org/10.7554/eLife.45457.011

interaction between tau and the CRL4$^{CRBN}$ E3 ligase may also occur endogenously in human neurons, as has been reported in tissue from mouse brain (*David et al., 2002*; *Del Prete et al., 2016*). Notably, P-tau high-molecular-weight species detected by PHF-1 (see 'flow through' of W:PHF-1, *Figure 4A*) were not captured by co-IP, which might be caused by bead-antibody alteration of epitope detection or simply low pull-down efficiency of these species in these conditions. Probing with an ubiquitin antibody (W:Ubi-1, *Figure 4A*) showed an increase in ubiquitinated proteins by QC-01–175 ± carfilzomib treatment, with the same migration as tau (~50 kDa), consistent with a QC-01–175-

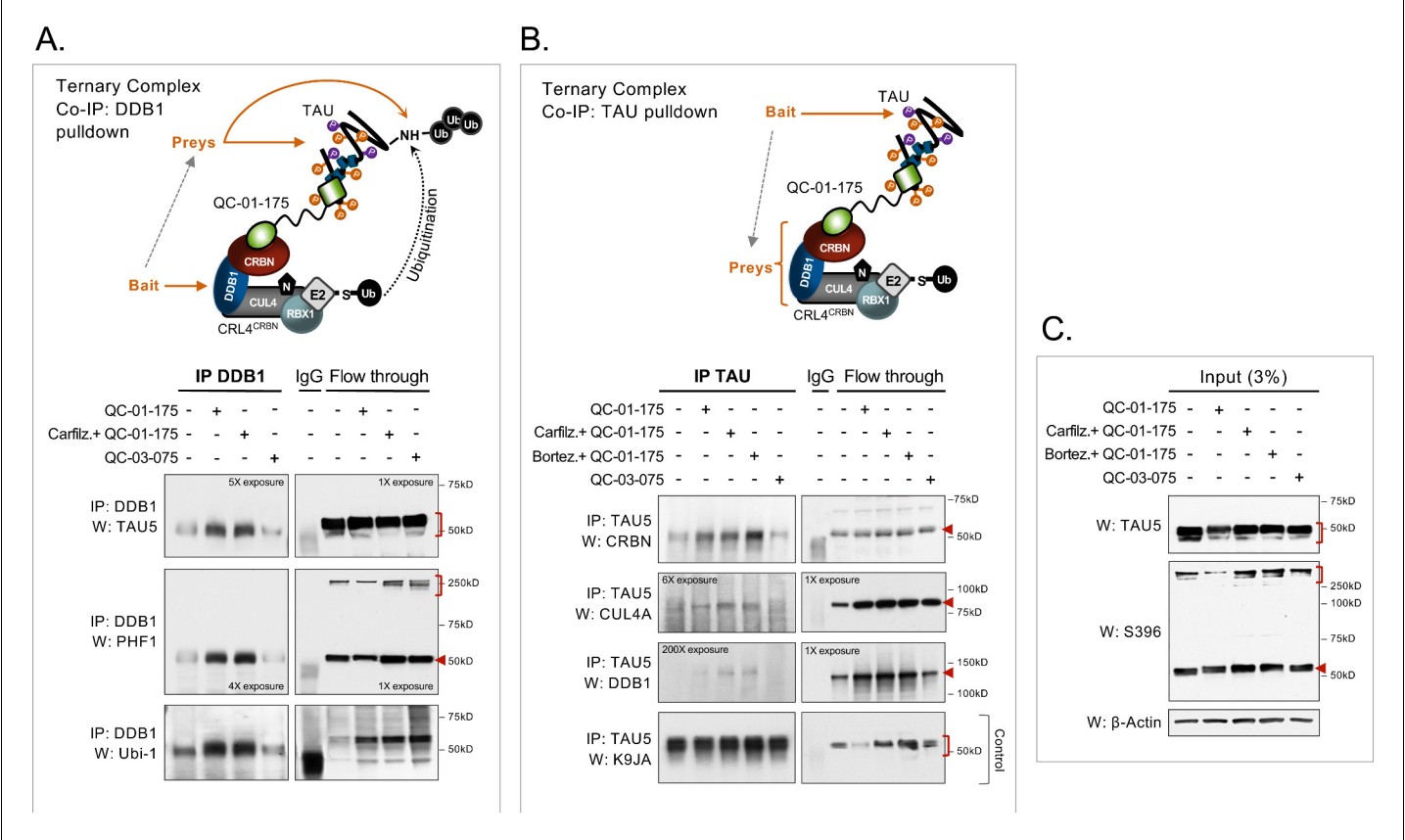

**Figure 4.** Demonstration of ternary complex formation in A152T neurons upon QC-01–175 treatment, by co-IP and western blot analysis. Neurons (6-week differentiated) were treated for 4 hr with 1 μM QC-01–175 ± 30 min pre-treatment with proteasome inhibitors (carfilzomib or bortezomib at 5 μM), with the goal of capturing maximum molecular interactions at 4 hr and halting tau clearance. QC-03–075 is a negative control for CRBN binding. (**A**) Co-IP by DDB1 pulldown and detection of tau in the complex by probing for total tau (TAU5), P-tau$^{S396/S404}$ (PHF-1), and ubiquitinated proteins (Ubi-1). (**B**) Co-IP by tau pulldown (TAU5) and detection of CRL4$^{CRBN}$ subunits CRBN, CUL4A and DDB1. Western blot of total tau (K9JA) was used as a control. (**C**) Control western blot analysis with 3% (10 μg) of IP input confirms the effect of QC-01–175 ± proteasome inhibitors on tau and P-tau S396. Red arrows and brackets indicate the predicted bands for each immunoprobed protein (*n* = 3).

DOI: https://doi.org/10.7554/eLife.45457.012

mediated increase in tau ubiquitination. Conversely, immunoprecipitation of tau (TAU5 antibody) and western blotting analysis showed co-IP with CRBN in QC-01–175 treated samples ± carfilzomib or bortezomib (*Figure 4B*). Again, a low level of CRBN-tau interaction was seen for vehicle and negative control QC-03–075-treated samples. CUL4A and DDB1 were also detected by western blotting, particularly in carfilzomib- and bortezomib-treated samples, but at much lower intensity (*Figure 4B*). Total tau K9JA western blotting was used as a control of tau IP. In both assays, comparison with flow-through control western blotting (notice film exposures, *Figure 4A,B*) shows that co-IP'ed proteins were a relatively low fraction of the total cellular amount, which was expected given the assay conditions and the transient nature of E3-ligase:substrate complexes. Differences in antibody sensitivities may also affect detection of each component of CRL4$^{CRBN}$ in *Figure 4B*. IP input-only western blotting analysis confirmed QC-01–175 degradation of tau and P-tau, as well as no effect on tau levels by the vehicle, negative control, or in carfilzomib/bortezomib pre-treated samples (*Figure 4C*). These results support ternary complex formation between tau, degrader and CRL4$^{CRBN}$ followed by increased tau ubiquitination, at the 4 hr time point, providing further evidence of the mechanism of tau clearance by QC-01–175 through E3 ligase function and proteasome degradation.

## Kinetics of tau degradation

In order to establish the kinetics of tau degradation, we treated A152T (*Figure 5*, *Figure 5—figure supplement 1A–C*) and P301L (*Figure 5—figure supplement 1F–H*) neurons for 4 or 8 hr, and compared tau degradation to the 24 hr treatment time-point using tau ELISA. For A152T neurons, at the 4 hr time-point we observed a sharp hook effect (*Huang and Dixit, 2016*) (*Figure 5—figure supplement 1A*), with maximum clearance of tau and P-tau$^{S396}$ ($D_{max, 4h}$ = 80%) at 100 nM (*Figure 5A–B*), constant 50% degradation at 500 nM and 1 µM (*Figure 5C*), and little effect on tau levels at 10 µM QC-01–175 (*Figure 5D*). After 8 hr, the hook effect lessened (*Figure 5—figure supplement 1B*), and $D_{max, 8h}$ of 50% tau was observed at all concentrations above 100 nM (*Figure 5A–D*). By comparison, at the 24 hr time-point tau and P-tau$^{S396}$ levels were further reduced in a concentration-dependent manner to a $D_{max, 24h}$ of 70% at 10 µM (*Figure 5—figure supplement 1C*). The 4 hr effect of QC-01–175 on A152T neurons was corroborated by western blotting analysis (*Figure 5—figure supplement 1D,E*). In P310L neurons, there was more variability between the two tau species at the 4 hr time-point (*Figure 5—figure supplement 1F*). For total tau, we observed a hook effect with $D_{max, 4h}$ occurring at 100 nM, whereas for P-tau$^{S396}$ maximal degradation was observed at 500 nM (both $D_{max, 4h}$ = 80%). However, by 8 hr robust degradation of tau and P-tau$^{S396}$ (40–50%) was achieved at all concentrations above 500 nM, and $D_{max, 8h}$ was further increased to 60–70% by 24 hr (*Figure 5—figure supplement 1G,H*).

## Evaluation of degrader specificity in neurons

To identify potential off-targets of the QC-01–175 degrader in the A152T neuronal proteome, we employed multiplexed MS-based proteomics to measure changes in protein abundance following a 4 hr treatment. Short treatments, such as this, enable some control over transcriptional and secondary effects that are often a consequence of prolonged drug exposure. Protein abundances were measured using tandem mass tag (TMT) isobaric labels, enabling the quantification of treatment effects with QC-01–175 (active degrader), QC-03–075 (inactive control), and QC-01–175 + MLN4924 (degradation rescue), relative to vehicle control. Whole proteome analysis resulted in the

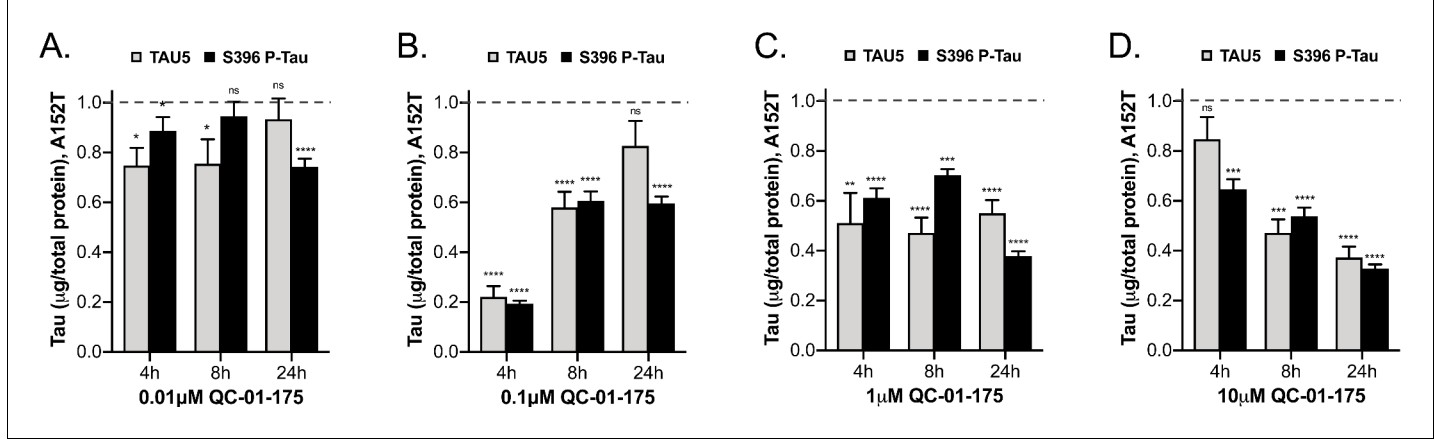

**Figure 5.** Comparative analysis of the effect of QC-01–175 at (**A**) 0.01 µM, (**B**) 0.1 µM, (**C**) 1 µM, and (**D**) 10 µM after 4 hr, 8 hr or 24 hr of treatment. Graph bars represent mean levels of total tau (TAU5) and S396 P-tau protein measured by ELISA, normalized to total µg of protein and to vehicle-treated samples ± SEM ($n$ = 3), in A152T 6-week differentiated neurons. Student T-test for each dose/time is relative to vehicle-treated tau levels [ns]p> 0.05, *p<0.05, **p<0.01, ***p<0.001, ****p<0.0001. *Figure 5—figure supplement 1* shows concentration effect curves for QC-01–175 at 4 hr and 8 hr, for all doses tested in A152T and P301L neurons; as well as the 4 hr effect seen by western blot. *Figure 5—source data 1* includes all values plotted in the main Figure and supplement. The following figure supplement is available for *Figure 5*.

DOI: https://doi.org/10.7554/eLife.45457.013

The following source data and figure supplement are available for figure 5:

**Source data 1.** Numerical description and statistics for data presented in *Figure 5* and respective supplement 1.
DOI: https://doi.org/10.7554/eLife.45457.014
**Figure supplement 1.** Degrader concentration and time effect on tau, in A152T and P301L neurons.
DOI: https://doi.org/10.7554/eLife.45457.015

identification and quantification of >8000 proteins with summed signal:noise >200 and >2 unique peptides. Upon 4 hr treatment with 1 µM QC-01–175, the only significant changes observed comprise the validated immune-modulatory drug (IMiD) targets ZFP91, ZNF653 and ZNF827 (*Donovan et al., 2018*), while no QC-01–175 specific off-targets were observed (*Figure 6A*). Degradation of these $C_2H_2$ zinc finger transcription factors was expected as an off-target response to the CRBN-binding module of QC-01–175 (pomalidomide, *Figure 1C*). Conversely, the negative control QC-03–075 (1 µM) revealed no effect on these proteins, confirming the ablation of CRBN-binding activity (*Figure 6B*). The degradation rescue experiment, consisting of pre-treatment with the neddylation inhibitor MLN4924 (inhibits activation of all NEDD8-dependent CRL) (*Lan et al., 2016*) and QC-01–175, showed a complete rescue of IMiD target degradation, confirming the neddylation and hence cullin-dependence for target degradation (*Figure 6C*). These results confirmed that QC-01–175 has minimal off-target activity. We were unable to reliably quantify tau by proteomics since the solubilization procedure necessary to dissolve insoluble tau species is incompatible with our pipeline.

## QC-01–175 rescues tau-mediated stress vulnerability of FTD neurons

In FTD patient-derived neuronal cell models, accumulation of tau and P-tau of reduced solubility is coupled to increased cellular vulnerability to specific forms of stress (*Silva et al., 2016*). One of these stressors is the proteotoxicity caused by the highly aggregation-prone peptide Aβ(1-42), which promotes a concentration-dependent loss of viability in A152T and P301L neurons (*Figure 7A*). This effect on viability is not seen in WT control neurons and can be rescued by CRISPR/Cas9-mediated *MAPT* knockout (KO) in A152T neurons (*Figure 7A*) (*Silva et al., 2016*). We tested whether QC-01–175 could similarly protect against tau-mediated stress vulnerability and loss of viability. A152T neurons were pre-treated with either 5 µM QC-01–175 or 5 µM QC-03–075 (*Figure 7B* b), or vehicle alone (DMSO, *Figure 7B* c) for 8 hr, followed by addition of 10 µM Aβ(1-42) stressor for an additional 16 hr (*Figure 7B* b, c). Vehicle- and degrader-only treated samples were also included as controls (*Figure 7B* a). Vehicle pre-treated neurons followed by Aβ(1-42) treatment loss 50% viability at the 24 hr time-point (*Figure 7C*). QC-01–175 pre-treatment rescued viability to 90% of vehicle-alone control, almost phenocopying the 100% viability rescue observed in *MAPT* KO neurons (*Figure 7C*).

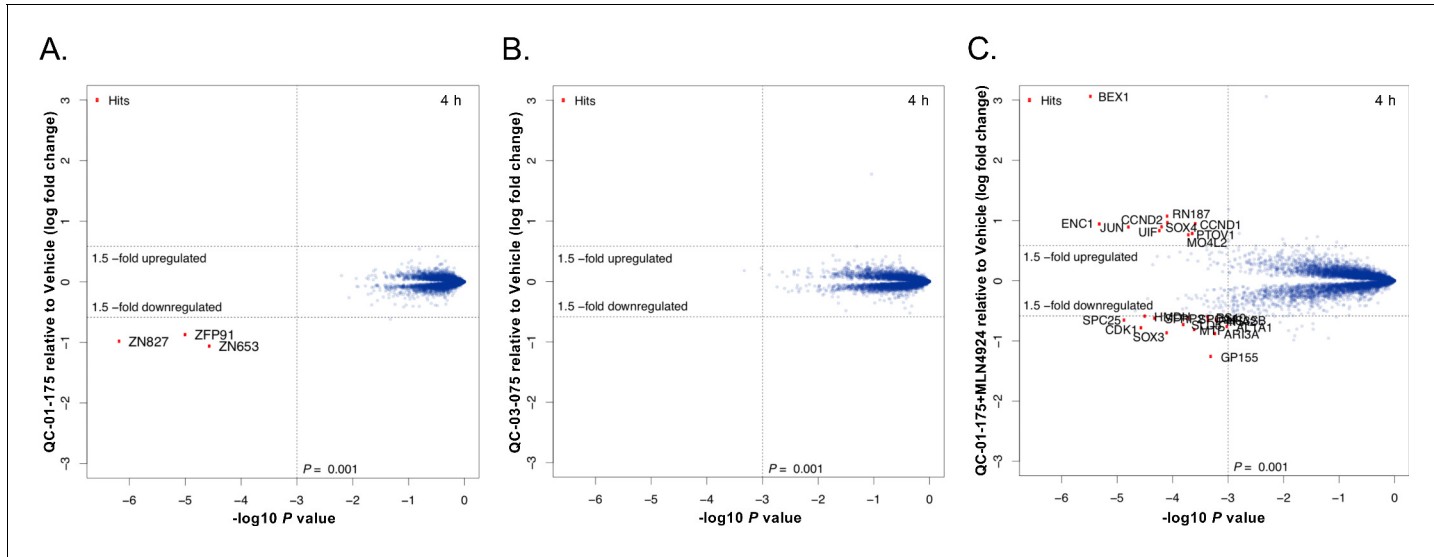

**Figure 6.** Mass spectrometry-based proteomics to quantify the effect of QC-01–175 treatment on the proteome of A152T neurons. 6-week differentiated neurons were treated for 4 hr with (**A**) 1 µM of QC-01–175, (**B**) 1 µM of the negative control QC-03–075, or (**C**) 10 µM MLN4924 (NAE inhibitor, 30 min pre-treatment) and 1 µM of QC-01–175. Upon degrader QC-01–175 treatment (**A**), three off-targets were detected within statistical significance, which all belong to known IMiD targets, an effect rescued by the negative control (**B**) or inhibition of neddylation by MLN4924 (**C**). Significant hits were assessed by moderated t-test as implemented in the limma package (*Ritchie et al., 2015*), with the log2 fold change shown on the y-axis, and negative log10 P values on the x-axis (n = 3 for treatment with DMSO, QC-01–175, and QC-03–075, and n = 2 for QC-01–175 + MLN4924).
DOI: https://doi.org/10.7554/eLife.45457.016

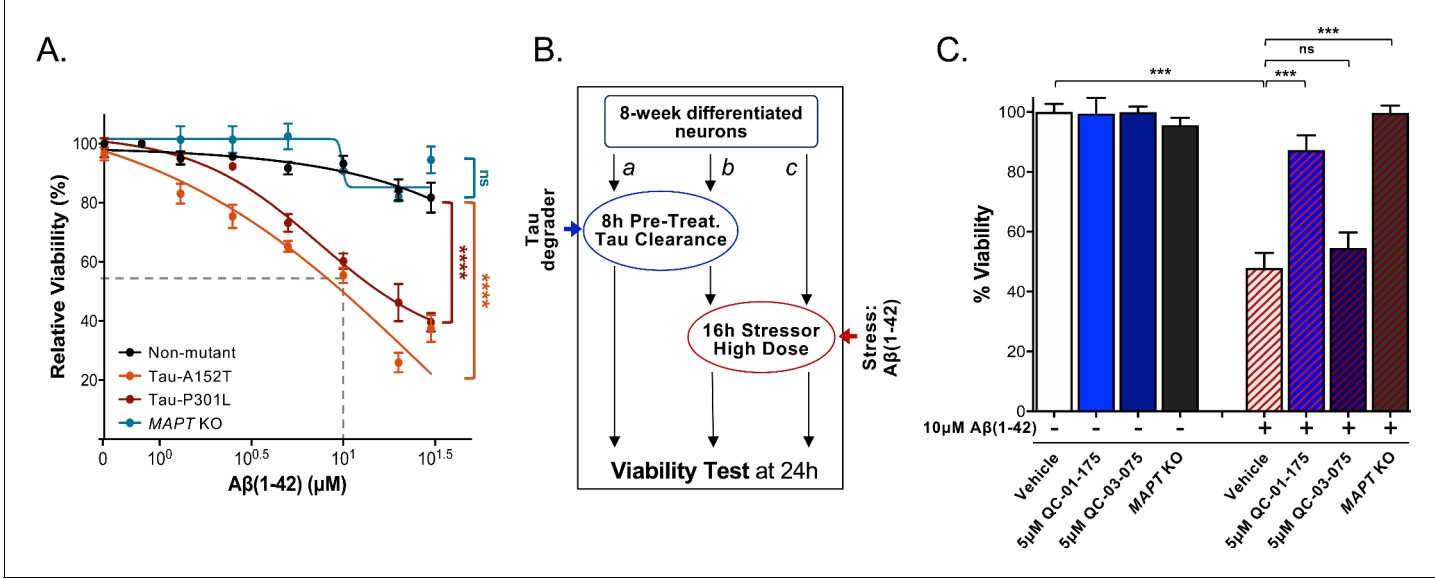

**Figure 7.** QC-01–175 treatment rescued stress vulnerability of A152T neurons. (**A**) Aβ(1-42) proteotoxicity causes concentration- and genotype-dependent loss of neuronal vulnerability, affecting preferentially A152T and P301L neurons, with a rescue by *MAPT* KO. Data points represent mean viability relative to vehicle-treated neurons (100%)±SEM ($n \geq 3$); two-way ANOVA statistical analysis relative to non-mutant control-1 neurons (black curve, 8330–8-RC1), ****$p<0.0001$, [ns]$P > 0.05$. (**B**) Assay overview to measure effect of the stressor Aβ(1-42) on neuronal viability (**c**) and potential rescue by pre-treatment with QC-01–175 (**b**). Effect of 24 hr treatment with QC-01–175 alone was also tested (**a**). (**C**) QC-01–175 (light blue-stripe bar) but not the negative control QC-03–075 (dark blue-stripe bar), rescued viability loss caused by Aβ(1-42) (white-stripe bar) in A152T neurons differentiated for 8 weeks, in a comparable manner to genetic *MAPT* knockout (black-stripe bar). Graph bars represent mean % viability ± SD, relative to vehicle-treated (white) neurons. T-test ***$p \leq 0.001$; [ns]$p>0.05$ ($n = 3$). *Figure 7—source data 1* includes all values plotted in main *Figure 7A and C*.
DOI: https://doi.org/10.7554/eLife.45457.017

The following source data is available for figure 7:

**Source data 1.** Numerical description and statistics for data presented.
DOI: https://doi.org/10.7554/eLife.45457.018

The negative control QC-03–075 had little effect on the toxicity of Aβ(1-42). As expected, 24 hr treatment with either QC-01–175 or negative control alone caused no change in neuronal viability (*Figure 7C*, left blue bars).

In conclusion, the tau degrader QC-01–175 triggers proteasome-dependent tau and P-tau clearance in FTD patient-derived neurons, which is able to rescue tau-mediated toxicity in biologically relevant assays of stress vulnerability.

## Discussion

In this study, we established that disease-relevant forms of tau and P-tau in FTD patient-derived neuronal cell models are amenable to targeted degradation and characterize the tool compound QC-01–175 as an effective tau degrader. We developed a BLI in vitro assay to assess QC-01–175 binding to recombinant tau species relative to T807. Despite detecting binding, QC-01–175 and T807 showed micromolar affinities to immobilized, soluble protein, independent of the presence of a tau variant. *E. coli*-expressed and purified recombinant human tau has distinct properties that are not reflective of intracellular protein conformations, which may be key to the recognition, binding and degradation of tau by QC-01–175 in the neuronal context. This is a limitation of current heterologous cell culture systems and in vitro measurements that do not recapitulate endogenous cellular tau properties. This highlights the importance of assaying disease-relevant tau species and an urgent need for improved biophysical binding assays. Introducing patient-specific neuronal models that recapitulate *early* aspects of tau pathology ex vivo in a drug discovery pipeline is unique to this study. Human stem cell-derived neurons allow access to physiologically relevant cell types and protein complexes, offering the advantage of modeling pathogenic tau species without the need for

overexpression of heterologous genes, in the context of the patient genomic background (*Dolmetsch and Geschwind, 2011*; *Silva et al., 2017*; *Wan et al., 2015*; *Chesselet and Carmichael, 2012*). QC-01–175 was able to promote tau clearance in FTD neurons expressing tau-A152T or tau-P301L, in a concentration-dependent manner, and subsequently rescued tau-mediated neuronal stress vulnerability. The effect of QC-01–175 in A152T and P301L heterozygous neurons led on average to more than 70% and 60% clearance of tau, respectively, when only about 50% of tau expressed has the variant. This is informative regarding the species of tau targeted by the degrader, raising the hypothesis that some forms of non-mutant tau in FTD neurons are also misfolded and targeted for degradation. Remarkably, the degrader preferentially targeted tau species from FTD neurons, with minimal effect on tau from unaffected WT control neurons. This indicates that it may be possible to develop tau-degrader therapies that selectively act in FTD-affected neurons, and on disease-associated forms of tau, independent of genotype, and spare normal tau protein within the affected regions of the brain.

Variation in QC-01–175-mediated tau degradation dose profiles between A152T and P301L neurons (*Figure 2C* and *Figure 2—figure supplement 2A*) points to differences in tau species originated by the two tau variants, and possibly also differences in proteostasis for the two patient-derived cell models, which affects UPS function. This aspect will impact the clinical translational potential of any degrader targeting tau to the proteasome, which has been demonstrated to be affected by neurodegeneration and aging itself (*Deger et al., 2015*; *Keck et al., 2003*; *Saez and Vilchez, 2014*). This, however, does not preclude the development of targeted degraders as therapeutic strategies for early stage disease, or as a co-therapeutic with proteasome activity chemical enhancers (*Myeku and Duff, 2018*). Notably, co-IP results (*Figure 4*) suggest that there is a basal level of interaction between tau and the CRL4$^{CRBN}$ E3 ligase in A152T neurons. Interestingly, Del Prete *et al.* reported that tau could be a natural substrate of CRBN through a complex involving the amyloid precursor protein (APP), mutations in which are known to cause AD (*Del Prete et al., 2016*). The co-IP experiment data we present here indicates that in FTD there may also be some basal tau: CRBN association, and thus, rather than targeting a 'neo-substrate', QC-01–175 may be promoting efficacy of a physiological process, increasing the probability of successful tau degradation in vivo. Further work is needed to investigate this hypothesis.

Important challenges still remain for translation of our promising initial findings into a therapeutic. From a pharmacology standpoint, QC-01–175 is a relatively large and flexible molecule compared to most brain-active drugs and may suffer from fast metabolism and poor brain penetration. Tuning pharmacology for in vivo applications is an ongoing challenge in the targeted protein degradation field, but progress is being made (*Gu et al., 2018*). If this tuning proves intractable, intrathecal administration could be considered. Clinical biomarkers for target engagement and efficacy evaluation will also be critical. Since QC-01–175 occupies the binding site of a PET ligand, it might be possible to use PET to measure target engagement in vivo. However, because degraders can act catalytically, their pharmacodynamics may influence and complicate PET readings. Further evaluation of QC-01–175 off-target binding will also be crucial. The degrader can recognize and downregulate other proteins as shown here by mass spectrometry global proteome analysis (*Figure 6A*), where three $C_2H_2$ zinc finger transcription factors were downregulated by a 4 hr treatment with QC-01–175. This off-target activity likely arises from the CRBN-binding module of QC-01–175 (pomalidomide, *Figure 1C*), which has been reported to induce degradation of multiple members of the $C_2H_2$ zinc finger protein family through a 'molecular glue' type mechanism (*Sievers et al., 2018*). Since these are lesser-known transcription factors, the cellular implications of their degradation are not yet fully understood. Using what is known about IMiD-based degrader design, efforts to design next generation tau degraders will involve tuning the attachment chemistry to replace the aniline nitrogen, and altering the degrader linker composition to remove these IMiD off-target effects (*Nowak et al., 2018*; *Dobrovolsky et al., 2019*; *Jiang et al., 2019*), thus creating a tau-selective degrader. These efforts are currently underway in our laboratory. However, QC-01–175 may also have modulatory effect on function/activity of other proteins without leading to degradation. Here, we tested and excluded the known MAO inhibitory effect of T807, but one could speculate that other enzymes (e.g. kinases) could be affected in vivo. Therefore, a detailed profiling of off-target binding interactions is a necessary future investigative effort. Finally, since it is now possible to routinely establish patient specific iPSC-derived neuronal cultures, an interesting strategy for clinical trial success might be to screen individual patients for responsiveness using cellular models.

In conclusion, here we have established a neurodegeneration-specific workflow for development of targeted degraders for aggregation-prone proteins. Our workflow leveraged existing PET probe binders in degrader design, and well-characterized patient-derived neuronal models to assay against clinically relevant protein species and phenotypes. These efforts yielded QC-01–175, a valuable new research tool for the study of tau-mediated events in human tauopathies. PET-tracer-based hetero-bifunctional degrader design and patient-derived cellular models offer an unprecedented advantage for the targeting of pathologically relevant tau species, relative to in vitro and heterologous expression systems. Additionally, we anticipate that future advances in understanding the relationship between tau post-translational modifications and adopted conformations in disease, as well as visualizing and modeling interactions between tau and various PET tracers, will aid in the knowledge-based design of next generation targeted degraders. Importantly, our approach may be generalizable to other proteinopathies where high-quality PET tracers and cellular models are available. In this context, our study suggests that small-molecule-mediated protein degradation represents a promising strategy to advance our understanding of human neurodegenerative disease and translate those insights into targeted therapies.

# Materials and methods

**Key resources table**

| Reagent type (species) or resource | Designation | Source or reference | Identifiers | Additional information |
|---|---|---|---|---|
| Cell line (*H. sapiens*) | 8330–8-RC1 | *Silva et al. (2016)* Stem Cell Reports. | | *Figure 2—source data 1*. Human iPSC-derived NPC line, non-mutant tau. Original fibroblasts GM08330 from Coriell Institute for Medical Research. |
| Cell line (*H. sapiens*) | MGH2069-RC1 | *Seo et al. (2017)* J. Neuroscience. *Manuscript in preparation.* | | *Figure 2—source data 1*. Human iPSC-derived NPC line, non-mutant tau. Original fibroblasts MGH-2069 from Massachusetts General Hospital Frontotemporal Dementia Clinic, Massachusetts General Hospital Neurodegeneration Repository. |
| Cell line (*H. sapiens*) | CTR2-L17-RC2 | *Almeida et al., 2012* Cell Reports. Silva *et al.* (2016) Stem Cell Reports | | *Figure 2—source data 1*. Human iPSC-derived NPC line, non-mutant tau. |
| Cell line (*H. sapiens*) | FTD19-L5-RC6 | *Silva et al. (2016)* Stem Cell Reports | | *Figure 2—source data 1*. Human iPSC-derived NPC line, tau-A152T (NCBI RefSeq NM_001123066; rs143624519). |
| Cell line (*H. sapiens*) | FTD19-L5-RC6;*MAPT-KO* | *Silva et al. (2016)* Stem Cell Reports | | *Figure 2—source data 1*. Human iPSC-derived NPC FTD19-L5-RC6 line, CRISPR/Cas9-engineered *MAPT* knockout. |
| Cell line (*H. sapiens*) | MGH2046-RC1 | *Seo et al. (2017)* J. Neuroscience. *Manuscript in preparation.* | | *Figure 2—source data 1*. Human iPSC-derived NPC line, tau-P301L (NCBI RefSeq NM_001123066; rs63751273). Original fibroblasts MGH-2046 from Massachusetts General Hospital Frontotemporal Dementia Clinic, Massachusetts General Hospital Neurodegeneration Repository. |

*Continued on next page*

Continued

| Reagent type (species) or resource | Designation | Source or reference | Identifiers | Additional information |
|---|---|---|---|---|
| Antibody | TAU5 | Invitrogen | Cat. AHB0042 RRID:AB_2536235 | WB 1:1000 |
| Antibody | TAU5 | AbCam | Cat. ab80579 RRID:AB_1603723 | Co-IP |
| Antibody | Tau K9JA | DAKO, Agilent | Cat. A002401-2 | IF 1:1000, WB 1:10,000 |
| Antibody | P-Tau S396 | Invitrogen | Cat. 44752G RRID:AB_1502108 | WB 1:1000 |
| Antibody | P-Tau PHF-1 | Dr. Peter Davies | Albert Einstein College of Medicine, NY | IF 1:400 |
| Antibody | MAP2 | Chemicon, Millipore | Cat. AB5543 RRID:AB_571049 | IF 1:1000 |
| Antibody | DDB1 | AbCam | Cat. ab109027 RRID:AB_10859111 | WB 1:50,000/Co-IP |
| Antibody | CUL4A | Cell Signaling Technology | Cat. 2699 RRID:AB_2086563 | WB 1:1000 |
| Antibody | CRBN | ProteinTech | Cat. 11435–1-AP RRID:AB_2085739 | WB 1:500 |
| Antibody | Ubiquitin, Ubi-1 | Millipore | Cat. MAB1510 RRID:AB_2180556 | WB 1:500 |
| Antibody | β-Actin | Sigma-Aldrich | Cat. A1978 RRID:AB_476692 | WB 1:10,000 |
| Antibody | GAPDH | AbCam | Cat. ab8245 RRID:AB_2107448 | WB 1:5000 |
| Antibody | AlexaFluor-488 2° antibody | Life Technologies | Cat. A11039 RRID:AB_142924 | IF 1:500 |
| Antibody | AlexaFluor-594 2° antibody | Life Technologies | Cat. A11012 RRID:AB_141359 | IF 1:500 |
| Antibody | AlexaFluor-594 2° antibody | Life Technologies | Cat. A11032 RRID:AB_141672 | IF 1:500 |
| Antibody | Anti-mouse IgG, HRP-linked | Cell Signaling Technology | Cat. 7076S RRID:AB_330924 | Western blotting, 1:4000 |
| Antibody | Anti-rabbit IgG, HRP-linked | Cell Signaling Technology | Cat. 7074S RRID:AB_2099233 | Western blotting, 1:4000 |
| Antibody | Hoechst 33342 | Invitrogen | Cat. H3570 | IF Nuclear stain, 1:1000 |
| Peptide, Recombinant Protein | Tau-441(WT), Biotinylated | SignalChem | Cat. T08-54BN Lot. H2681-10 | Human recombinant protein expressed in *E. coli* cells. |
| Peptide, Recombinant Protein | Tau-441(A152T) Protein | SignalChem | Cat. T08-56VN Lot. B2157-7 | Human recombinant protein expressed in *E. coli* cells, tag-free. Accession no. P10636-8. |
| Peptide, Recombinant Protein | Tau-441(P301L) Protein | SignalChem | Cat. T08-56FN Lot. O917-2 | Human recombinant protein expressed in *E. coli* cells, tag-free. Accession no. P10636-8. |
| Peptide, Recombinant Protein | Aβ(1-42) | Enzo Lifesciences | Cat. ALX-151–002 | CAS No. 107761-42-2 |
| Commercial Assay, Kit | EZ-Link NHS-PEG$_4$-Biotinylation Kit | Thermo Fisher Scientific | Cat. 21455 | |
| Commercial Assay, Kit | MAO-Glo Assay Kit | Promega | Cat. V1401 | |
| Commercial Assay, Kit | Pierce BCA Protein Assay Kit | Thermo Fisher Scientific | Cat. 23227 | |
| Commercial Assay, Kit | Human Total Tau ELISA | Invitrogen | Cat. KHB0041 | |

*Continued*

| Reagent type (species) or resource | Designation | Source or reference | Identifiers | Additional information |
|---|---|---|---|---|
| Commercial Assay, Kit | P-Tau[pS396] Human ELISA | Invitrogen | Cat. KHB7031 | |
| Commercial Assay, Kit | ELISA Compatible Lysis Buffer | Invitrogen | Cat. FNN0011 | |
| Commercial Assay, Kit | Immunoprecipitation Kit Dynabeads Protein G | Novex, Life Technologies | Cat. 10007D | |
| Commercial Assay, Kit | Pierce IP Lysis Buffer | Thermo Fisher Scientific | Cat. 87787 | |
| Commercial Assay, Kit | Tandem mass tag (TMT) reagents | Thermo Fisher Scientific | Cat. A34807 | |
| Commercial Assay, Kit | AlamarBlue Cell Viability Reagent | Thermo Fisher Scientific | Cat. DAL1025 | |
| Chemical Compound, Drug | T807 (AV-1451) | MedChem Express | Cat. HY-101184 | CAS No. 1415379-56-4 |
| Chemical Compound, Drug | T807 core scaffold | This paper | (Intermediate 10) | Methods, Synthetic methods general protocols. *Figure 1C*. |
| Chemical Compound, Drug | Pomalidomide | Sigma Aldrich | Cat. P0018 | CAS No. 19171-19-8 |
| Chemical Compound, Drug | Lenalidomide | Sigma Aldrich | Cat. 901558 | CAS No. 191732-72-6 |
| Chemical Compound, Drug | QC-01–175 | This paper | | Methods, Synthetic methods general protocols. *Figure 1C*. |
| Chemical Compound, Drug | QC-03–075 | This paper | | Methods, Synthetic methods general protocols. *Figure 1C*. |
| Chemical Compound, Drug | MLN4924 | MedChem Express | Cat. HY-70062 | CAS No. 905579-51-3 |
| Chemical Compound, Drug | Bafilomycin A1 | Enzo LifeSciences | Cat. BML-CM110 | CAS No. 88899-55-2 |
| Chemical Compound, Drug | Carfilzomib | MedChem Express | Cat. HY-10455 | CAS No. 868540-17-4 |
| Chemical Compound, Drug | Bortezomib | Selleckchem | Cat. S1013 | CAS No. 179324-69-7 |
| Chemical Compound, Drug | PE859 | MedChem Express | Cat. HY-12662 | CAS No. 1402727-29-0 |
| Chemical Compound, Drug | Parnate (Tranylcypromine) | Sigma-Aldrich | Cat. P8511 | CAS No. 1986-47-6 |
| Chemical Compound, Drug | Protease inhibitor cocktail | Roche | Cat. 04 693 124 001 | |
| Chemical Compound, Drug | Phosphatase inhibitor cocktail 2 | Sigma-Aldrich | Cat. P5726 | |
| Software, Algorithm | Data Acquisition HT 11.0 | ForteBio (www.fortebio.com/octet-software.html) | | Version 11 (BLI Analysis and $K_D$ calculation) |
| Software, Algorithm | Adobe Photoshop CS5 | Adobe Photoshop (www.adobe.com/Photoshop) | | Version 12.0.4 (Histogram function, western blots densitometry) |
| Software, Algorithm | GraphPad Prism | GraphPad Prism (www.graphpad.com) | | Version 8 |
| Software, Algorithm | Proteome Discoverer 2.2 | Thermo Fisher Scientific | RRID:SCR_014477 | Version 2.2 |
| Software, Algorithm | R framework | Team RCR: A Language and Environment for Statistical Computing http://www.R-project.org/; accessed Nov. 1, 2017 | | R Version 3.5.1 – Feather Spray |
| Software, Algorithm | Statistical Analysis Limma Package (R framework) | Bioconductor | | *Ritchie et al. (2015)* Nucleic Acids Res. |

*Continued on next page*

*Continued*

| Reagent type (species) or resource | Designation | Source or reference | Identifiers | Additional information |
|---|---|---|---|---|
| Other | Octet Red384 Instrument | ForteBio | | https://www.fortebio.com/octet-red384.html |
| Other | IN Cell Analyzer 6000 Cell Imaging System | GE Healthcare Life Sciences | | |
| Other | EnVision Multilabel Plate Reader | Perkin Elmer | | |
| Other | HPLC | Waters 2489/2545 | | |
| Other | UPLC | Waters Aquity I UPLC | | |
| Other | HPLC | Agilent 1260 Infinity II LC System | | |
| Other | Orbitrap Fusion Lumos mass spectrometer | Thermo Fisher Scientific | IQLAAEGAAPFADBMBHQ | |
| Other | Proxeon EASY-nLC 1200 LC pump | Thermo Fisher Scientific | LC140 | |
| Other | EasySpray ES803 75 μm inner diameter microcapillary column | Thermo Fisher Scientific | ES803 | |

## General protocols

Unless otherwise noted, reagents and solvents were obtained from commercial suppliers and were used without further purification. $^1$H NMR spectra were recorded on 500 MHz Bruker Avance III spectrometer, and chemical shifts are reported in parts per million (ppm, δ) downfield from tetramethylsilane (TMS). Coupling constants (J) are reported in Hz. Spin multiplicities are described as s (singlet), br (broad singlet), d (doublet), t (triplet), q (quartet), and m (multiplet). Mass spectra were obtained on a Waters Acquity UPLC. Preparative HPLC was performed on a Waters Sunfire C18 column (19 mm ×50 mm, 5 μM) using a gradient of 15–95% methanol in water containing 0.05% trifluoroacetic acid (TFA) over 22 min (28 min run time) at a flow rate of 20 mL/min. Assayed compounds were isolated and tested as TFA salts. Purities of assayed compounds were in all cases greater than 95%, as determined by reverse-phase HPLC analysis.

**Scheme 1.** Synthesis route for the tau degrader QC-01-175.
DOI: https://doi.org/10.7554/eLife.45457.019

## Synthetic procedures

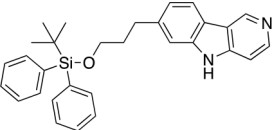

**Chemical structure 1.** 3-(4-(4-nitropyridin-3-yl)phenyl)propan-1-ol (3)
DOI: https://doi.org/10.7554/eLife.45457.020

A solution of **1** (1.65 g, 6.29 mmol), **2** (1.16 g, 5.72 mmol), $Na_2CO_3$ (1.52 g, 14.3 mmol) and Pd $(PPh_3)_4$ (330 mg, 0.286 mmol) in 1,4-dioxane (40 mL) and $H_2O$ (10 mL) was stirred at 110°C for 16 hr before it was quenched with $NH_4Cl$ (sat. aq., 100 mL). The resulting mixture was extracted with $CH_2Cl_2$ (3 × 80 mL), the combined organic phases were dried over anhydrous $Na_2SO_4$ and concentrated under reduced pressure. The residue was purified by flash column chromatography (0–10% MeOH in DCM) to give **3** (950 mg, 3.68 mmol, 64% yield). [1]H NMR (500 MHz, DMSO-$d_6$) δ 8.81 (d, J = 5.3 Hz, 1H), 8.78 (s, 1H), 7.89 (d, J = 5.2 Hz, 1H), 7.59–7.41 (m, 4H), 4.47 (t, J = 5.1 Hz, 1H), 3.39 (td, J = 6.4, 5.0 Hz, 2H), 2.65–2.59 (m, 2H), 1.73–1.65 (m, 2H). [13]C NMR (126 MHz, DMSO) δ 153.05, 151.11, 143.80, 131.99, 131.96, 131.91, 130.68, 129.45, 129.40, 129.28, 129.21, 129.12, 129.10, 128.71, 128.57, 117.09, 60.54, 34.48, 31.83, 31.80. MS (ESI) *m/z* 259 (M + H)[+]. Expected mass from chemical formula $C_{14}H_{14}N_2O_3$: 258.3 Da.

**Chemical structure 2.** 3-(4-(3-((*tert*-butyldiphenylsilyl)oxy)propyl)phenyl)-4-nitropyridine (4)
DOI: https://doi.org/10.7554/eLife.45457.021

To a stirred solution of **3** (950 mg, 3.68 mmol) in $CH_2Cl_2$ (35 mL) at 25°C was added imidazole (751 mg, 11.04 mmol) and TBDPSCl (2.01 g, 7.36 mmol). After stirring at this temperature for 3 hr, the reaction was quenched with $NH_4Cl$ (sat. aq., 100 mL). The resulting mixture was extracted with $CH_2Cl_2$ (2 × 50 mL), the combined organic phases were washed with brine (100 mL), dried over anhydrous $Na_2SO_4$ and concentrated under reduced pressure. The residue was purified by flash column chromatography (0–5% MeOH in DCM) to give **4** (1.62 g, 3.26 mmol, 89% yield). [1]H NMR (500 MHz, DMSO-$d_6$) δ 8.90 (d, J = 5.3 Hz, 1H), 8.85 (s, 1H), 7.99 (d, J = 5.2 Hz, 1H), 7.66–7.61 (m, 4H), 7.50–7.41 (m, 6H), 7.35–7.29 (m, 4H), 3.69 (t, J = 6.2 Hz, 2H), 2.74 (dd, J = 8.6, 6.7 Hz, 2H), 1.92–1.83 (m, 2H), 1.02 (s, 9H). [13]C NMR (126 MHz, DMSO) δ 154.54, 153.05, 151.19, 143.25, 136.83, 135.49, 133.71, 130.80, 130.27, 129.63, 129.43, 128.63, 128.35, 117.14, 63.14, 33.88, 31.48, 27.02, 19.15. MS (ESI) *m/z* 497 (M + H)[+]. Expected mass from chemical formula C30H32N2O3Si: 496.68 Da.

**Chemical structure 3.** 7-(3-((*tert*-butyldiphenylsilyl)oxy)propyl)-5H-pyrido[4,3-b]indole (5)
DOI: https://doi.org/10.7554/eLife.45457.022

A solution of **4** (1.62 g, 3.26 mmol) in $P(OEt)_3$ (20 mL) was stirred at 110°C for 3 hr before it was concentrated under reduced pressure. The residue was purified by flash column chromatography (0–10% MeOH in DCM) to give **5** (1.32 g, 2.84 mmol, 87% yield). [1]H NMR (500 MHz, DMSO-$d_6$) δ 11.65 (s, 1H), 9.30 (s, 1H), 8.41 (dd, J = 5.7, 1.7 Hz, 1H), 8.09 (d, J = 7.9 Hz, 1H), 7.63–7.60 (m, 4H), 7.48–7.36 (m, 8H), 7.08 (dt, J = 8.0, 1.5 Hz, 1H), 3.68 (t, J = 6.2 Hz, 2H), 2.85 (t, J = 7.6 Hz, 2H), 1.96–1.88 (m, 2H), 1.02 (s, 9H). [13]C NMR (126 MHz, DMSO) δ 144.35, 144.16, 142.58, 141.14, 140.49, 135.47, 133.74, 130.22, 128.29, 121.33, 120.85, 119.95, 119.21, 111.35, 106.78, 63.11, 34.57, 32.36, 27.16, 19.27. MS (ESI) *m/z* 465 (M + H)[+]. Expected mass from chemical formula C30H32N2O3Si: 464.23 Da.

**Chemical structure 4.** *tert*-butyl 7-(3-((*tert*-butyldiphenylsilyl)oxy)propyl)-*5H*-pyrido[4,3-*b*]indole-5-carboxylate (6)
DOI: https://doi.org/10.7554/eLife.45457.023

To a stirred solution of **5** (1.32 g, 2.84 mmol) and DMAP (213 mg, 1.75 mmol) in $CH_2Cl_2$ (30 mL) at 25°C was added $Et_3N$ (1.06 g, 10.47 mmol) and (Boc)$_2$O (1.52 g, 6.98 mmol). After stirring at this temperature for 2 hr, the reaction was quenched with $NH_4Cl$ (sat. aq., 100 mL). The resulting mixture was extracted with $CH_2Cl_2$ (2 × 50 mL), the combined organic phases were washed with brine (100 mL), dried over anhydrous $Na_2SO_4$ and concentrated under reduced pressure. The residue was purified by flash column chromatography to give **6** (1.42 g, 2.52 mmol, 89% yield). $^1$H NMR (500 MHz, DMSO-$d_6$) δ 9.35 (d, *J* = 0.9 Hz, 1H), 8.58 (d, *J* = 5.7 Hz, 1H), 8.12 (d, *J* = 7.9 Hz, 1H), 8.09 (d, *J* = 1.3 Hz, 1H), 8.05 (dd, *J* = 5.7, 0.9 Hz, 1H), 7.61–7.57 (m, 4H), 7.43–7.34 (m, 6H), 7.25 (dd, *J* = 7.9, 1.5 Hz, 1H), 3.67 (t, *J* = 6.1 Hz, 2H), 2.86 (t, *J* = 7.5 Hz, 2H), 1.91 (dq, *J* = 8.5, 6.3 Hz, 2H), 1.68 (s, 9H), 0.99 (s, 9H). $^{13}$C NMR (126 MHz, DMSO) δ 150.29, 147.22, 142.96, 142.68, 138.42, 135.43, 133.66, 130.18, 128.24, 128.21, 124.88, 121.77, 121.40, 120.77, 115.99, 110.99, 85.39, 62.99, 34.32, 32.60, 28.14, 28.13, 27.10, 19.22. MS (ESI) *m/z* 565 (M + H)$^+$. Expected mass from chemical formula $C_{35}H_{40}N_2O_3Si$: 564.28 Da.

**Chemical structure 5.** *tert*-butyl 7-(3-hydroxypropyl)-*5H*-pyrido[4,3-*b*]indole-5-carboxylate (7)
DOI: https://doi.org/10.7554/eLife.45457.024

To a stirred solution of **6** (1.42 g, 2.52 mmol) in THF (25 mL) at 0°C was added TBAF (1.0 M in THF, 3.8 mL, 3.8 mmol) dropwise. After stirring at this temperature for 2 hr, the reaction was quenched with acetic acid (0.2 mL). The mixture was concentrated under reduced pressure. The residue was purified by flash column chromatography (0–20% MeOH in DCM) to give **7** (670 mg, 2.05 mmol, 81% yield). $^1$H NMR (500 MHz, DMSO-$d_6$) δ 9.35 (s, 1H), 8.58 (d, *J* = 5.7 Hz, 1H), 8.14 (d, *J* = 7.9 Hz, 1H), 8.12 (s, 1H), 8.04 (d, *J* = 5.7 Hz, 1H), 7.30 (dd, *J* = 7.9, 1.3 Hz, 1H), 3.47 (t, *J* = 6.4 Hz, 2H), 2.81 (dd, *J* = 8.8, 6.7 Hz, 2H), 1.86–1.77 (m, 2H), 1.72 (s, 9H). $^{13}$C NMR (126 MHz, DMSO) δ 149.89, 146.72, 142.85, 142.48, 138.03, 124.53, 121.33, 120.84, 120.29, 115.42, 110.57, 85.02, 60.04, 39.52, 34.51, 32.40, 27.73, 21.06. MS (ESI) *m/z* 327 (M + H)$^+$. Expected mass from chemical formula $C_{19}H_{22}N_2O_3$: 326.16 Da.

**Chemical structure 6.** 3-(*5H*-pyrido[4,3-*b*]indol-7-yl)propanoic acid (10)
DOI: https://doi.org/10.7554/eLife.45457.025

To a stirred solution of DMSO (1.30 g, 16.6 mmol) in $CH_2Cl_2$ (7 mL) at –78°C was added oxalyl chloride (783 mg, 6.16 mmol) in $CH_2Cl_2$ (6 mL) dropwise. After stirring at this temperature for 0.5 hr, a solution of **7** (670 mg, 2.05 mmol) in $CH_2Cl_2$ (6 mL) was added dropwise. The mixture was stirred at this temperature for 2 hr followed by the addition of $Et_3N$ (1.035 g, 10.25 mmol) dropwise. The reaction mixture was slowly warmed to 0°C over 1 hr and was quenched with $NH_4Cl$ (sat. aq., 30 mL). The resulting mixture was extracted with $CH_2Cl_2$ (2 × 30 mL), the combined organic phases were washed with brine (100 mL), dried over anhydrous $Na_2SO_4$ and concentrated under reduced pressure. The residue was used in the next step without further purification.

To a stirred solution of the above residue in THF (10 mL), *t*-BuOH (5 mL) and $H_2O$ (5 mL) at 25°C was added $NaH_2PO_4 \cdot H_2O$ (2.55 g, 18.5 mmol), 2-methyl-butene (5 mL) and sodium chlorite (1.64 g, 18.5 mmol). After stirring at this temperature for 2 hr, the reaction was diluted with $H_2O$ (50 mL). The resulting mixture was extracted with $CH_2Cl_2$ (2 × 50 mL), the combined organic phases were

dried over anhydrous $Na_2SO_4$ and concentrated under reduced pressure. The residue was used in the next step without further purification.

A solution of the above mixture in $CH_2Cl_2$ (12 mL) and TFA (6 mL) was stirred at 25°C for 12 hr before it was concentrated under reduced pressure. The residue was dissolved in NaOH (0.5 M, aq., 20 mL) and was extracted with $CH_2Cl_2$ (4 × 15 mL). The water phase was added HCl (aq.,1.0 M) dropwise to adjust the pH to 6–7. The resulting mixture was extracted with $CHCl_3$/$^i$PrOH (4/1, 3 × 30 mL), the combined organic phases were dried over anhydrous $Na_2SO_4$ and concentrated under reduced pressure to afford **10** (357 mg, 1.48 mmol, 73% yield over three steps) as a pure compound. $^1$H NMR (500 MHz, Methanol-$d_4$) δ 9.22 (s, 1H), 8.37 (s, 1H), 8.10 (d, $J$ = 8.0 Hz, 1H), 7.58 (d, $J$ = 6.0 Hz, 1H), 7.48 (s, 1H), 7.27 (dd, $J$ = 8.1, 1.4 Hz, 1H), 3.12 (t, $J$ = 7.7 Hz, 2H), 2.63 (t, $J$ = 7.8 Hz, 2H). $^{13}$C NMR (126 MHz, MeOD) δ 172.25, 142.51, 139.68, 138.42, 122.01, 120.84, 120.39, 118.91, 110.98, 106.62, 32.19, 29.36. MS (ESI) $m/z$ 241 (M + H)$^+$. Expected mass from chemical formula $C_{14}H_{12}N_2O_2$: 240.09 Da.

**Chemical structure 7.** QC-01-175

DOI: https://doi.org/10.7554/eLife.45457.026

To a stirred solution of carboxylic acid **10** (6.8 mg, 0.02 mmol), EDCI (11.5 mg, 0.06 mmol), DMAP (2.2 mg, 0.02 mmol) and DIPEA (14.3 mg, 0.12 mmol) in $CH_2Cl_2$ (0.6 mL) at 25°C was added the corresponding primary amine (12.1 mg, 0.03 mmol). The resulting reaction mixture was stirred at this temperature for 4 hr, and then concentrated under reduced pressure. The residue was purified by Reverse-Phase HPLC to give **QC-01–175** (9.7 mg, 0.0134 mmol, 67% yield). $^1$H NMR (500 MHz, Methanol-$d_4$) δ 9.44 (s, 1H), 8.47 (d, $J$ = 6.7 Hz, 1H), 8.21 (d, $J$ = 8.1 Hz, 1H), 7.88 (d, $J$ = 6.7 Hz, 1H), 7.56 (s, 1H), 7.43 (dd, $J$ = 8.5, 7.1 Hz, 1H), 7.36 (dd, $J$ = 8.1, 1.3 Hz, 1H), 6.96 (d, $J$ = 8.5 Hz, 1H), 6.91 (d, $J$ = 7.0 Hz, 1H), 5.02 (dd, $J$ = 12.8, 5.5 Hz, 1H), 3.76–3.69 (m, 2H), 3.67–3.64 (m, 6H), 3.60–3.55 (m, 3H), 3.53 (dd, $J$ = 6.0, 3.2 Hz, 2H), 3.47 (t, $J$ = 5.4 Hz, 2H), 3.40 (t, $J$ = 5.2 Hz, 2H), 3.13 (t, $J$ = 7.5 Hz, 2H), 2.58 (t, $J$ = 7.5 Hz, 2H). $^{13}$C NMR (126 MHz, MeOD) δ 146.59, 143.73, 135.71, 134.14, 133.82, 123.70, 121.45, 116.71, 112.08, 110.53, 107.77, 99.98, 70.14, 69.83, 69.17, 69.13, 48.77, 41.79, 39.02, 31.95, 30.77, 22.41. MS (ESI) $m/z$ 627 (M + H)$^+$. Expected mass from chemical formula $C_{33}H_{34}N_6O_7$: 626.25 Da.

**Chemical structure 8.** QC-03-075: Was prepared according to Scheme 1.

DOI: https://doi.org/10.7554/eLife.45457.027

$^1$H NMR (500 MHz, DMSO-$d_6$) δ 13.01 (s, 1H), 9.66 (s, 1H), 8.61 (d, $J$ = 6.7 Hz, 1H), 8.31 (d, $J$ = 8.1 Hz, 1H), 7.95 (d, $J$ = 6.6 Hz, 1H), 7.90 (t, $J$ = 5.7 Hz, 1H), 7.82 (d, $J$ = 2.9 Hz, 1H), 7.58 (s, 1H), 7.56–7.49 (m, 1H), 7.34 (d, $J$ = 8.1 Hz, 1H), 7.07 (d, $J$ = 8.6 Hz, 1H), 6.97 (d, $J$ = 7.0 Hz, 1H), 6.54 (t, $J$ = 5.8 Hz, 1H), 4.50 (dd, $J$ = 11.9, 6.4 Hz, 1H), 3.57 (t, $J$ = 5.5 Hz, 2H), 3.51 (dd, $J$ = 6.0, 3.4 Hz, 2H), 3.46 (t, $J$ = 3.5 Hz, 2H), 3.42 (d, $J$ = 5.8 Hz, 1H), 3.19 (q, $J$ = 5.4 Hz, 4H), 3.03 (t, $J$ = 7.6 Hz, 2H), 2.47 (d, $J$ = 7.7 Hz, 4H), 2.19 (qd, $J$ = 12.3, 4.4 Hz, 1H), 2.01–1.78 (m, 3H). $^{13}$C NMR (126 MHz, DMSO) δ 146.95, 135.34, 123.75, 122.31, 117.58, 112.67, 110.90, 108.62, 84.75, 76.26, 70.11, 69.66, 69.32, 49.04, 49.04, 42.14, 41.87, 37.53, 32.07, 26.33, 22.19. MS (ESI) $m/z$ 613 (M + H)$^+$. Expected mass from chemical formula $C_{33}H_{36}N_6O_6$: 612.27 Da.

## Bio-layer interferometry (BLI) biosensor assay

BLI was performed in the Octet Red384 instrument (ForteBio, Fremont CA, USA) with 1X PBS and 0.01% Brij-35 as the assay buffer. Recombinant human tau proteins (Tau-441(WT), Tau-441(A152T) and Tau-441(P301L), 2N4R isoform) were purchased from SignalChem (Richmond, British Columbia). Tau-441(WT), Tau-441(A152T) and Tau-441(P301L) were labeled with biotin using the EZ-LinkNHS-PEG4-Biotinylation Kit (Thermo Scientific, Waltham, MA), and excess biotin reagent was removed using a spin desalting column as per manufacturer's instruction. The biotinylated protein samples for BLI were purified in 1X phosphate buffered saline (PBS). Streptavidin (SA) sensors were used to measure the biophysical interaction between the small molecules and biotinylated tau. Prior to the initiation of BLI, the SA sensors were soaked by dipping in 200 μL of assay buffer in a 96-well Greiner Bio-One Black flat bottom plate (Greiner, Monroe NC, USA Cat.655209). The assay was performed in 80 μL volume in Greiner Bio-One 384-well black flat bottom PP plates (Greiner, Cat.781209) with an initial baseline step, followed by loading of 250 nM biotinylated tau protein. The small molecules samples and recombinant proteins were arranged in the 384-well plate according to a plate map recommended for the eight-channel mode kinetic analysis, where the sensors move from low to high concentration of small molecule samples. Subsequent steps included a second baseline (120 s), followed by association (240 s) and dissociation (240 s) cycles. All the sensors were loaded with biotinylated tau protein, and two sensors were loaded with appropriate concentration of DMSO (comparable to small molecule samples) in assay buffer to be used as reference. Data analysis was performed using Data Acquisition HT 11.0 software following reference subtraction (an average of two sensors with DMSO) using a 1:1 binding model and an individual fit of each replicate. The equilibrium dissociation constant ($K_D$) was estimated using data at equilibrium from each available small molecule concentration with steady-state analysis. The instrument manufacturer (Fortebio, article #137) recommends the steady state option for analyzing interactions that are either low affinity or very fast on-and-off affinity rates. For steady-state analysis of 'R equilibrium' ($R_{eq}$) was fitted according to 1:1 binding model with the equation Response = $(R_{max} \times Conc.)/(K_D + Conc.)$. When $R_{eq}$ option is selected, Fortebio's software calculates affinity constants based on the $R_{eq}$ values determined from the curve fits. In the steady-state analysis, $R_{eq}$ is plotted against the small molecule concentration to infer the $R_{max}$. $K_D$ is estimated as the concentration where 50% of $R_{max}$ is achieved. As per the instrument manufacturer's instructions, if all curves have reached equilibrium, these two sets of values correspond to 'Response,' and $R_{eq}$ values should match. For T807 and QC-01–175, the values of 'Response' and $R_{eq}$ match as evidenced by the raw data as well as a value of $R_{max}/R_{eq}$ (%) close to 1.

## Monoamine oxidase assay

The Monoamine oxidase (MAO) activity measurements were performed using a MAO-Glo assay kit (Promega, Madison, WI), in a 384-well plate (Proxiplate 384 Plus, Perkin Elmer, Waltham, MA) with the miniaturization of final assay volume to 20 μL, as per the manufacturer's instructions. In brief, MAO reactions were terminated after a 60-min enzymatic reaction, by the addition of reconstituted Luciferin Detection Reagent. Following a 20-min incubation with the detection reagent, the luminescence was measured using a multi-label plate reader (Envision, Perkin Elmer). $IC_{50}$ values were estimated by fitting percentage inhibition vs. compound concentration in a variable slope (four parameter) fit.

## Human neuronal cell culture and compound treatment

Cells from individuals carrying the tau risk variant A152T (c.1407G > A; NCBI RefSeq NM_001123066, rs143624519), the autosomal dominant mutation P301L (c.C1907T; NCBI NM_001123066, rs63751273), or age-matched non-mutant WT tau were employed in this study (*Figure 2—source data 1*). Dermal fibroblasts from the tau-P301L carrier and the healthy individual were generated from a skin biopsy from subjects within the MGH Frontotemporal Disorders Unit as part of the MGH Neurodegeneration Repository. Approval for human subjects' work was obtained under a Partners/MGH-approved IRB Protocol (#2010P001611/MGH). Fibroblasts were reprogrammed into iPSCs, which were subsequently converted into cortical-enriched neural progenitor cells (NPCs) and differentiated into neuronal cells as previously described (*Almeida et al., 2012*; *Seo et al., 2017*; *Silva et al., 2016*; *Sheridan et al., 2011*) (*Silva MC, Manuscript in preparation*,

*2019*). Briefly, cells were cultured in 6-well (Fisher Scientific Corning, Pittsburgh, PA) or 96-well (Fisher Scientific Corning) plates coated with poly-ornithine (20 μg/mL in water, Sigma, St. Louis, MO) and laminin (5 μg/mL in PBS, Sigma) (POL-coated), in DMEM/F12-B27 media [70% DMEM (Gibco, Carlsbad, CA), 30% Ham's-F12 (Fisher Scientific Corning), 2% B27 (Gibco), 1% penicillin-streptomycin (Gibco)]. Media was supplemented with EGF (20 ng/mL, Sigma), FGF (20 ng/mL, Stemgent, Cambridge, MA) and heparin (5 μg/mL, Sigma), to promote NPC proliferation and expansion. The growth factors were withdrawn to promote neural differentiation for 6 to 8 weeks, with half media change two times per week. Cell lines' identity was authenticated by SANGER sequencing, karyotyping and ACGH analysis at different stages of culture maintenance from iPSC to NPC. All working cell lines tested negative for mycoplasma contamination.

Compound treatment in six-well plates was performed in 2 mL media volume by removing 1 mL of conditioned media from the culture and adding 1 mL of new media pre-mixed with the compound at the appropriate 2X concentration, followed by incubation at 37°C for the designated period of time. Compound treatment in 96-well plates was performed in 100 μL media volume by adding compound directly to each well, followed by incubation at 37°C. When testing the effect of drug pre-treatment, the first compound was added as described above for the period of time needed, and then QC-01–175 was added directly onto the media without media exchange.

## Antibodies and respective commercial information

Total tau TAU5 for western blotting (Invitrogen, Rockford, IL), total tau TAU5 for IP (AbCam, Cambridge, MA), total tau K9JA for IF (Dako/Agilent, Santa Clara, CA), P-tau Ser396 (Invitrogen), P-tau PHF-1 (kindly provided by Dr. Peter Davies, Albert Einstein College of Medicine, NY). Neuronal marker MAP2 (Chemicon/Millipore). CRL4$^{CRBN}$ E3 Ligase components DDB1 (AbCam), CUL4A (Cell Signaling Technology), and CRBN (ProteinTech, Rosemont, IL). Anti-Ubiquitin, clone Ubi-1 (Millipore, Darmstadt, Germany). Internal controls GAPDH (Abcam) and β-Actin (Sigma). Nuclear stain Hoechst 33342 (Invitrogen).

## Cell lysis and western blotting analysis

Neurons differentiated in six-well plates for 6 weeks were washed and collected in PBS, lysed in RIPA buffer (Boston Bio-Products, Boston, MA) supplemented with 2% SDS (Sigma), protease inhibitors cocktail (Roche Complete Mini tablets, Mannheim, Germany), and phosphatase inhibitors cocktail (Sigma), followed by water sonication (Bransonic Ultrasonic Baths, Thomas Scientific, Danbury, CT) and 20,000 *g* centrifugation for 20 min. Supernatants were transferred to new tubes and total protein concentration was quantified with the Pierce BCA Protein Assay Kit (ThermoFisher Scientific, Carlsbad, CA). Human recombinant tau protein ladder was purchased from Sigma. Electrophoresis were performed with the Novex NuPAGE SDS-PAGE Gel System (Invitrogen, Carlsbad, CA), by running 10 μg of total protein (pre-boiled in SDS-DTT loading buffer, NEB, Ipswich, MA) on pre-cast SDS-PAGE. Gels were transferred onto PVDF membranes (EMD Millipore) using standard procedures. Membranes were blocked in 5% BSA (Sigma) in Tris-buffered saline with Tween-20 (TBST/Biorad, Hercules, CA) for 2 hr, incubated overnight with primary antibody (see antibody section) at 4°C, followed by corresponding HRP-linked secondary antibody incubation (Cell Signaling Technology, Danvers, MA). Blots were developed with SuperSignal West Pico Chemiluminescent Substrate (ThermoFisher) according to manufacturer's instructions and exposed to autoradiographic films (LabScientific by ThermoFischer) that, in turn, were scanned on an Epson Perfection V800 Photo Scanner. Protein bands densitometry (pixel mean intensity) was measured with the Adobe Photoshop CS5 Histogram function and normalized to the respective internal control (β-Actin or GAPDH) band.

## Human tau ELISA

Neuronal cells differentiated in six-well plate format for 6 weeks, were collected as described for western blotting, and lysed in ELISA-compatible buffer (Invitrogen FNN0011), supplemented with 1 mM PMSF, protease (Roche) and phosphatase (Sigma) inhibitors cocktails, for 30 min on ice with quick vortexing. Lysates were clarified by centrifugation at 13,000 rpm at 4°C for 10 min. The clear lysates were then transferred to new microfuge tubes and total protein concentration was quantified with the Pierce BCA Protein Assay Kit (ThermoFisher Scientific). ELISA assays were performed according to manufacturer instructions: Human Total Tau ELISA (Invitrogen Kit KHB0041/KHB0042)

and P-Tau[pS396] Human ELISA Kit (Invitrogen Kit KHB7031). Data was plotted using GraphPad Prism version 8.0.

## Immunofluorescence assay

A152T NPCs were plated and differentiated in 96-well black clear-bottom plates (Corning) POL-coated for 6 weeks. Neurons were fixed with 4% (v/v) formaldehyde-PBS (Tousimis, Rockville, MD) for 20 min, washed in PBS (Corning), incubated in blocking/permeabilization buffer [10 mg/mL BSA (Sigma), 0.05% (v/v) Tween-20 (Biorad), 2% (v/v) goat serum, 0.1% Triton X-100 (Biorad), 92% (v/v) PBS] for 2 hr, followed by overnight incubation with primary antibodies (see antibody section), PBS washed, and then incubated with the corresponding AlexaFluor conjugated secondary antibodies (Life Technologies, Carlsbad, CA). Image acquisition was done with the IN Cell Analyzer 6000 Cell Imaging System (GE Healthcare Life Sciences, Marlborough, MA).

## Co-immunoprecipitation assays

NPCs were plated and differentiated for 6 weeks, as described above, in six-well POL-coated plates. For each pull-down experiment, three wells of treated neurons were PBS-washed and combined into a single pellet. To stabilize and detect complex formation, all steps were performed on ice or at 4°C. Co-IP was performed with the Immunoprecipitation Kit Dynabeads Protein G (Novex by Life Technologies/Thermo 10007D), according to manufacturer instructions. Briefly, cell lysis was achieved with ice cold Pierce IP Lysis Buffer (ThermoFisher Scientific) for 15 min at 4°C, followed by centrifugation at 10,000 $g$ for 10 min. The supernatant (*Input*) was transferred to a new tube and immediately aliquoted for the BCA assay to determine protein concentration. Antibodies, 10 μg of TAU5 and 5 μg of DDB1, were bound to Dynabeads (1.5 mg) per each condition to be tested (treatments). Immunoprecipitation of the target antigen (tau or DDB1) was performed by mixing protein lysate (200 μL sample corresponding to 300 μg of total protein) to each Dynabeads-antibody complex. All other steps followed the kit's protocol. Elution was accomplished with 20 μL Elution Buffer and 10 μL of SDS-sample buffer with DTT (New England Biolabs), followed by removal of the magnetic beads. A volume of 10 μL of each IP sample was loaded onto SDS-PAGE for western blotting as described above.

## Mass spectrometry global proteomics

### Sample preparation TMT LC-MS3 mass spectrometry

A152T neurons at 6 weeks of differentiation were treated with DMSO vehicle, 1 μM of degrader QC-01–175 or 1 μM negative control QC-03–075 in biological triplicates for 4 hr, or pre-treated for 30 min with 10 μM MLN4924 followed by 1 μM QC-01–175 addition for 3.5 hr, in biological duplicates. Neuronal cells were washed in PBS (Corning VWR, Radnor, PA) and collected at 3000 $g$ centrifugation. Lysis buffer (8 M Urea, 50 mM NaCl, 50 mM 4-(2hydroxyethyl)−1-piperazineethanesulfonic acid (EPPS) pH 8.5, protease and phosphatase inhibitors (Roche) were added to the cell pellets and homogenized by 20 passes through a 21 gauge (1.25 in. long) needle to achieve a cell lysate with a protein concentration between 0.25–2 mg/mL. A micro-BCA assay (Pierce) was used to determine final protein concentration in the cell lysates. 100 μg of protein for each sample were reduced and alkylated as previously described (*Donovan et al., 2018*). Proteins were precipitated using methanol/chloroform. In brief, four volumes of methanol were added to the cell lysate, followed by one volume of chloroform, and finally three volumes of water. The mixture was vortexed and centrifuged to separate the chloroform phase from the aqueous phase. The precipitated protein was washed with three volumes of methanol, centrifuged and the resulting washed precipitated protein was allowed to air dry. Precipitated protein was resuspended in 4 M Urea, 50 mM HEPES pH 7.4, followed by dilution to 1 M urea with the addition of 200 mM EPPS, pH 8. Proteins were first digested with LysC (1:50; enzyme:protein; Fisher Scientific) for 12 hr at room temperature. The LysC digestion was diluted to 0.5 M Urea with 200 mM EPPS pH eight followed by digestion with trypsin (1:50; enzyme:protein; Promega) for 6 hr at 37°C. Tandem mass tag (TMT) reagents (Thermo Fisher Scientific) were dissolved in anhydrous acetonitrile (ACN) according to manufacturer's instructions. Anhydrous ACN was added to each peptide sample to a final concentration of 30% v/v, and labeling was induced with the addition of TMT reagent to each sample at a ratio of 1:4 peptide:TMT label. The 11-plex labeling reactions were performed for 1.5 hr at room temperature and the reaction

quenched by the addition of hydroxylamine to a final concentration of 0.3% for 15 min at room temperature. The sample channels were combined at a 1:1:1:1:1:1:1:1:1:1:1 ratio, desalted using $C_{18}$ solid phase extraction cartridges (Waters, Milford, MA) and analyzed by LC-MS for channel ratio comparison. Samples were then combined using the adjusted volumes determined in the channel ratio analysis and dried down in a speed vacuum. The combined sample was then resuspended in 1% formic acid and acidified (pH 2–3) before being subjected to desalting with C18 SPE (Sep-Pak, Waters). Samples were then offline fractionated into 96 fractions by high-pH reverse-phase HPLC (Agilent LC1260, Santa Clara, CA) through an aeris peptide xb-c18 column (phenomenex) with mobile phase A containing 5% acetonitrile and 10 mM $NH_4HCO_3$ in LC-MS grade $H_2O$, and mobile phase B containing 90% acetonitrile and 10 mM $NH_4HCO_3$ in LC-MS grade $H_2O$ (both pH 8.0). The 96 resulting fractions were then pooled in a non-continuous manner into 24 fractions and these fractions were used for subsequent mass spectrometry analysis.

Data were collected using an Orbitrap Fusion Lumos mass spectrometer (ThermoFisher Scientific, San Jose, CA) coupled with a Proxeon EASY-nLC 1200 LC pump (ThermoFisher Scientific). Peptides were separated on an EasySpray ES803 75 µm inner diameter microcapillary column (ThermoFisher Scientific). Peptides were separated using a 190 min gradient of 6–27% acetonitrile in 1.0% formic acid with a flow rate of 350 nL/min. Each analysis used an MS3-based TMT method as described previously (*McAlister et al., 2014*). The data were acquired using a mass range of *m/z* 340–1350, resolution 120,000, AGC target $5 \times 10^5$, maximum injection time 100 ms, dynamic exclusion of 120 s for the peptide measurements in the Orbitrap. Data dependent MS2 spectra were acquired in the ion trap with a normalized collision energy (NCE) set at 35%, AGC target set to $1.8 \times 10^4$ and a maximum injection time of 120 ms. MS3 scans were acquired in the Orbitrap with HCD collision energy set to 55%, AGC target set to $2 \times 10^5$, maximum injection time of 150 ms, resolution at 50,000 and with a maximum synchronous precursor selection (SPS) precursors set to 10. The Advanced Peak Detection (APD) algorithm was disabled.

## LC-MS data analysis

Proteome Discoverer 2.2 (ThermoFisher Scientific) was used for .RAW file processing and controlling peptide and protein level false discovery rates, assembling proteins from peptides, and protein quantification from peptides. MS/MS spectra were searched against a Uniprot human database (September 2016) with both the forward and reverse sequences. Database search criteria are as follows: tryptic with two missed cleavages, a precursor mass tolerance of 20 ppm, fragment ion mass tolerance of 0.6 Da, static alkylation of cysteine (57.0211 Da), static TMT labeling of lysine residues and N-termini of peptides (229.163 Da), variable oxidation of methionine (15.9951 Da), variable phosphorylation of serine, threonine and tyrosine (79.966 Da) and variable acetylation (42.011 Da), Methionine-loss (131.040 Da) or methionine-loss + acetylation (83.030 Da) of the protein N-terminus. TMT reporter ion intensities were measured using a 0.003 Da window around the theoretical *m/z* for each reporter ion in the MS3 scan. Peptide spectral matches with poor quality MS3 spectra were excluded from quantitation (summed signal-to-noise across 11 channels < 200 and precursor isolation specificity < 0.5), and resulting data was filtered to only include proteins that had a minimum of two unique peptides identified. Reporter ion intensities were normalized and scaled using in-house scripts in the R framework (Team RCR: A Language and Environment for Statistical Computing http://www.R-project.org/; accessed Nov. 1, 2017). Statistical analysis was carried out using the limma package within the R framework (*Ritchie et al., 2015*).

## Neuronal stress and viability assay

Stress vulnerability assays were performed as previously described (*Silva et al., 2016*) (*Figure 7B*). NPCs were plated and differentiated in 96-well plate format, for eight weeks. Either QC-01–175, QC-03–075 or vehicle (DMSO) were added directly into the media (100 µL) to a final concentration of 5 µM, and incubated for 8 hr at 37°C. Then, each well was treated with either 10 µM of amyloid-beta(1-42) (Enzo Lifesciences, Farmingdale, NY), or vehicle alone, for an additional 16 hr incubation. At 24 hr, viability was measured with the Alamar Blue Cell viability reagent (Life Technologies), according to manufacturer instructions. Readings were done in the EnVision Multilabel Plate Reader (Perkin Elmer, Waltham, MA).

## Data availability

The compounds QC-01-175-1 and QC-03-075-1 are available from the Gray laboratory upon request. All mass spectrometry raw data is deposited and made available via the PRIDE archive under project accession number PXD012515.

# Acknowledgements

We wish to thank Dr. Kelly L Arnett and Harvard's Center for Macromolecular Interaction for advice regarding Biolayer Interferometry (BLI), and Dr. Peter Davies (Albert Einstein College of Medicine, NY) for kindly sharing the Tau PHF-1 antibody. Members of the Haggarty, Gray, Fischer laboratory along with Dr. Milka Kostic and the Tau Consortium Drug Discovery Group are thanked for helpful feedback on the experimental data and manuscript content.

# Additional information

### Competing interests

M Catarina Silva, Fleur M Ferguson, Quan Cai: is a co-inventor on a patent covering the molecules disclosed in this publication (WO/2019/014429). Bradford C Dickerson: is a consultant for Merck, Lilly, Biogen, and Piramal; and receives royalties from Oxford University Press, Cambridge University Press, and Elsevier. Eric S Fischer: is a SAB member and equity holder in C4 Therapeutics and a consultant to Novartis, AbbVie, and Pfizer. The Fischer lab receives or has received research funding from Novartis, Astellas and Deerfield. Nathanael S Gray: is a founder, science advisory board member (SAB) and equity holder in Gatekeeper, Syros, Petra, C4, B2S and Soltego. The Gray lab receives or has received research funding from Novartis, Takeda, Astellas, Taiho, Jansen, Kinogen, Her2llc, Deerfield and Sanofi. Co-inventor on a patent covering the molecules disclosed in this publication (WO/2019/014429). Stephen J Haggarty: is a member of the SAB and equity holder in Rodin Therapeutics, Psy Therapeutics, and Frequency Therapeutics. His laboratory has received funding from the Tau Consortium, F-Prime Biomedical Research Initiative, AstraZeneca, JW Pharmaceuticals and speaking fees from AstraZeneca, Amgen, Merck. Co-inventor on a patent covering the molecules disclosed in this publication (WO/2019/014429). The other authors declare that no competing interests exist.

### Funding

| Funder | Grant reference number | Author |
|---|---|---|
| National Institutes of Health | R21NS085487 | M Catarina Silva<br>Diane E Lucente<br>Bradford C Dickerson<br>Stephen J Haggarty |
| Tau Consortium | | M Catarina Silva<br>Ghata Nandi<br>Debasis Patnaik<br>Stephen J Haggarty |
| F-Prime Biomedical Research Initiative | | M Catarina Silva<br>Fleur M Ferguson<br>Ghata Nandi<br>Debasis Patnaik<br>Nathanael S Gray<br>Stephen J Haggarty |
| National Institutes of Health | R01CA214608 | Katherine A Donovan<br>Eric S Fischer |
| National Institutes of Health | R01CA218278 | Katherine A Donovan<br>Eric S Fischer |

The funders had no role in study design, data collection and interpretation, or the decision to submit the work for publication. Eric S Fischer is a Damon Runyon-Rachleff Innovator supported in part by the Damon Runyon Cancer Research Foundation (DRR-50-18).

## Author contributions

M Catarina Silva, Fleur M Ferguson, Katherine A Donovan, Conceptualization, Resources, Formal analysis, Investigation, Methodology, Writing—original draft, Writing—review and editing; Quan Cai, Conceptualization, Resources, Investigation, Methodology; Ghata Nandi, Debasis Patnaik, Resources, Formal analysis, Investigation; Tinghu Zhang, Hai-Tsang Huang, Resources, Investigation; Diane E Lucente, Resources, Investigation, Writing—review and editing; Bradford C Dickerson, Conceptualization, Resources, Investigation, Writing—review and editing; Timothy J Mitchison, Conceptualization, Writing—review and editing; Eric S Fischer, Conceptualization, Resources, Supervision, Methodology, Writing—review and editing; Nathanael S Gray, Stephen J Haggarty, Conceptualization, Resources, Supervision, Funding acquisition, Methodology, Project administration, Writing—review and editing

## Author ORCIDs

M Catarina Silva (iD) http://orcid.org/0000-0001-5421-6673
Fleur M Ferguson (iD) http://orcid.org/0000-0003-4091-7617
Katherine A Donovan (iD) http://orcid.org/0000-0002-8539-5106
Ghata Nandi (iD) http://orcid.org/0000-0001-7170-8709
Hai-Tsang Huang (iD) http://orcid.org/0000-0002-4244-2304
Eric S Fischer (iD) http://orcid.org/0000-0001-7337-6306
Nathanael S Gray (iD) https://orcid.org/0000-0001-5354-7403
Stephen J Haggarty (iD) http://orcid.org/0000-0002-7872-168X

## Decision letter and Author response

Decision letter https://doi.org/10.7554/eLife.45457.036
Author response https://doi.org/10.7554/eLife.45457.037

# Additional files

## Supplementary files

• Supplementary file 1. 1H NMR Spectra of QC-01-175.
DOI: https://doi.org/10.7554/eLife.45457.028

• Supplementary file 2. 1H NMR Spectra of QC-03-075.
DOI: https://doi.org/10.7554/eLife.45457.029

• Supplementary file 3. UPLC chromatogram and mass spectra of QC-01-175.
DOI: https://doi.org/10.7554/eLife.45457.030

• Supplementary file 4. UPLC chromatogram and mass spectra of QC-03-075.
DOI: https://doi.org/10.7554/eLife.45457.031

• Transparent reporting form
DOI: https://doi.org/10.7554/eLife.45457.032

## Data availability

Mass spectrometry global proteomics data is available via the PRIDE archive, under the project accession number PXD012515. Source data files have been provided for Figures 2, 5 and 7.

The following dataset was generated:

| Author(s) | Year | Dataset title | Dataset URL | Database and Identifier |
|---|---|---|---|---|
| Eric S Fischer | 2019 | Targeted Degradation of Aberrant Tau in Frontotemporal Dementia Patient-Derived Neuronal Cell | https://www.ebi.ac.uk/pride/archive/projects/PXD012515 | PRIDE, PXD012515 |

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
