## [Decision Letter]

Thank you for submitting your article "Targeted degradation of aberrant tau in frontotemporal dementia patient-derived neuronal cell models" for consideration by *eLife*. Your article has been reviewed by three peer reviewers, including Benjamin F Cravatt as the Reviewing Editor and Reviewer #1, and the evaluation has been overseen by Gary Westbrook as the Senior Editor. The reviewers have discussed the reviews with one another and the Reviewing Editor has drafted this decision to help you prepare a revised submission.

The reviewers and Editors agreed that your manuscript reports an interesting and important set of findings related to the chemically induced degradation of tau based on combining a PET tracer for this protein with an E3 ligase (cereblon)-directed ligand into a novel bifunctional probe. The data supporting the activity of the probe as an inducer of tau degradation in cells were recognized as convincing. The reviewers, however, requested an additional experiment to strengthen the manuscript. Specifically, we believe that the binding of the bifunctional probe QC-01-175 should be measured with wild type tau by BLI, as currently only mutant tau is measured. Because the compound has no effect on wild type tau in iPSCs (an interesting finding), it is important to understand whether this effect is due to (a) lack of binding to wild type tau or (b) lack of formation of the proper tau conformer in cells. This relatively straightforward experiment would seem to clarify the mechanistic possibilities. The reviewers also requested that some comments be made about the potential off-targets of the bifunctional probe, which would be important to investigate in future studies.

Summary of reviewer comments:

This is an interesting proof-of-principle study, which shows that PROTAC molecules can be directed to selectively degrade a misfolded protein, tau. Compared to other PROTACs studies, this seems to be the most challenging example, given the lack of tau structures and the disordered nature of the target. Towards this goal, the authors first synthesize a truncated form of the tau PET ligand, T807, and fuse it to the cereblon ligand, pomalidomide (QC-01-175). Importantly, they also create a key control (QC-03-075), unable to bind cereblon. Using BLI, they show that QC-01-175 binds to biotinylated A152T or P301L tau with a low μM affinity. Then, a series of cell-based experiments (Western blots, microscopy and ELISA) are used to show that QC-01-175 induces proteasomal degradation of mutated tau in differentiated, patient-derived iPSC neurons. The authors include many of the key controls that one expects in this type of study, including competition experiments and evidence of a hook effect. Finally, they show evidence that QC-01-175 is able to partially favor the ternary complex in cells by co-IP and that QC-01-175 has surprisingly few off target activities by mass spectrometry. Another highlight of the work is the transparent and welcome discussion of experimental variability.

Overall, the work has a number of interesting aspects worthy of publication. First, the fact that QC-01-175 induces degradation of A152T and P301L tau, but not wild type tau in control neurons is fascinating. Tau must, at least partially, adopt a structure that is recognized by T807 in the mutant cells – which is somewhat unexpected. Second, the selectivity of a T807-derived compound in cells (as shown by MS) is unexpected and interesting. Although there are some mechanistic details that remain mysterious and these compounds don't seem likely to have the BBB permeability to make them therapeutic leads, these discoveries more than compensate. Overall, this is a great manuscript and it should be of interest to *eLife* users. I would also venture that it will be a popular journal club entry, given both the technical advances and the mechanistic questions remaining.

Essential revisions:

Perhaps the biggest surprise in the work is that the T807-derived molecules does not reduce tau levels in the control neurons. This finding suggests that tau somehow fails to adopt a T807-binding competent structure in these cells. This aspect of the work could use a little more exploration. Does QC-10-175 bind to wild type tau fibrils by BLI?

---

## [Author Response]

[…] The reviewers, however, requested an additional experiment to strengthen the manuscript. Specifically, we believe that the binding of the bifunctional probe QC-01-175 should be measured with wild type tau by BLI, as currently only mutant tau is measured. Because the compound has no effect on wild type tau in iPSCs (an interesting finding), it is important to understand whether this effect is due to (a) lack of binding to wild type tau or (b) lack of formation of the proper tau conformer in cells. This relatively straightforward experiment would seem to clarify the mechanistic possibilities.Essential revisions:Perhaps the biggest surprise in the work is that the T807-derived molecules does not reduce tau levels in the control neurons. This finding suggests that tau somehow fails to adopt a T807-binding competent structure in these cells. This aspect of the work could use a little more exploration. Does QC-10-175 bind to wild type tau fibrils by BLI?

We thank the reviewers for the feedback on this experiment and for the opportunity to complement the data presented. We have now included BLI measurements of QC-01-175 in vitro binding to WT tau in Figure 1—figure supplement 1A (I-III) and Figure 1—figure supplement 1B (I-III). Results description is included in the second paragraph of the Results subsection “Design and in vitro testing of targeted tau degraders”.

We developed a BLI in vitro assay to determine whether QC-01-175, a derivative of T807 with known binding properties to pathological tau, would retain tau binding properties. This assay allowed us to show that QC-01-175 derivation from T807 did not dramatically alter binding to tau as the K_D_ values obtained were within the same order of magnitude (compare Figure 1—figure supplement 1B II-III to V-VI and VIII-IX). But, despite detecting binding, one cannot assume that in vitro and in vivo (or intracellular neuronal) tau conformations are comparable. *E. coli*-expressed tau certainly has different properties from endogenous neuronal expressed tau. Also, for this in vitro assay one assumes that the protein is immobilized in a soluble and monomeric form. Therefore, this BLI assay was not able to distinguish binding affinities of QC-01-175 to WT vs. variant forms of tau, which would be dependent on other factors such as conformation and post-translational modifications. We reason this explains why the results are different for QC-01-175 binding to all forms of tau in vitro vs. preferentially degrading mutant tau in our neuronal FTD-derived models. QC-01-175 binding to tau in our cellular models is shown by the CRBN co-IP assay results, indicating recognition is indeed preserved. Cleary this indicates limitations of using recombinant tau preparations in a BLI assay format and the urgent need for developing improved biophysical assays of physiologically relevant conformations of tau. This has now been added to the first paragraph of the Discussion section.

The reviewers also requested that some comments be made about the potential off-targets of the bifunctional probe, which would be important to investigate in future studies.

Potential off-target binding for the bifunctional molecule QC-01-175 could be mediated by either the tau-binding moiety, derived from T807, or by the CRBN ligand, in this case derived from pomalidomide (Figure 1C I). To address the first hypothesis, we tested the well-known off-target activity against monoamine oxidase-B (MAO-B) and monoamine oxidase-A (MAO-A) (Lemoine et al., 2018, Vermeiren et al., 2018) by implementing an in vitro MAO inhibition assay (relocated Figure 1—figure supplement 2), which revealed that QC-01-175 showed significantly reduced inhibition of MAO relative to T807. To address pomalidomide off-target engagement, as well as overall QC-01-175 effect on the global proteome, we employed multiplexed MS-based proteomics. Upon 4 h treatment with QC-01-175, the only significant changes observed comprise the validated immune-modulatory drug (IMiD) targets ZFP91, ZNF653 and ZNF827 (Donovan et al., 2018), while no QC-01-175 specific off-targets were observed (see Figure 6, and the Results subsection “Evaluation of degrader specificity in neurons”). We have also done this analysis upon 24 h treatment (not included) and the results were very similar with the same three IMiD targets showing the largest and only off-target significant effect (after tau). Future work to avoid IMiD target effect is now discussed in the third paragraph of the Discussion section.

Also, one cannot exclude other interactions by QC-01-175 that do not alter protein levels but modulate function and enzymatic activity. This is a very relevant and complex point that is brought up again in the Discussion section, where potential challenges for in vivo and clinical applications are discussed.